



# CICE on a C-grid: new momentum, stress, and transport schemes for CICEv6.5

Jean-François Lemieux[1], William H. Lipscomb[2], Anthony Craig[3], David A. Bailey[2], Elizabeth Hunke[4], Philippe Blain[5], Till A. S. Rasmussen[6], Mats Bentsen[7], Frédéric Dupont[5], David Hebert[8], and Richard Allard[8]

[1]Recherche en Prévision Numérique Environnementale, Environnement et Changement Climatique Dorval, Qc, Canada.
[2]Climate and Global Dynamics Laboratory, NSF National Center for Atmospheric Research, Boulder, CO, USA
[3]Contractor to Science and Technology Corporation, Seattle, WA, USA
[4]Los Alamos National Laboratory, Los Alamos, NM, USA
[5]Service Météorologique Canadien, Prévision numérique de l'environnement marin, Environnement et Changement Climatique Dorval, Qc, Canada.
[6]Danish Meteorological Institute, Copenhagen, Denmark
[7]NORCE Norwegian Research Centre AS and Bjerknes Centre for Climate Research, Bergen, Norway
[8]Naval Research Laboratory Stennis Space Center, Stennis, MS, USA

**Correspondence:** Jean-François Lemieux (jean-francois.lemieux@ec.gc.ca)

**Abstract.** This article presents the C-grid implementation of the CICE sea ice model, including the C-grid discretization of the momentum equation, the boundary conditions, and the modifications to the code required to use the incremental remapping transport scheme. To validate the new C-grid implementation, many numerical experiments were conducted and compared to the B-grid solutions. In idealized experiments, the standard advection method (incremental remapping with C-grid velocities

interpolated to the cell corners) leads to a checkerboard pattern. A modal analysis demonstrates that this computational noise originates from the spatial averaging of C-grid velocities at corners. The checkerboard pattern can be eliminated by adjusting the departure regions to match the divergence obtained from the solution of the momentum equation. We refer to this approach as the edge flux adjustment method. The C-grid discretization with edge flux adjustment allows transport in channels that are one grid cell wide—a capability that is not possible with the B-grid discretization nor with the C-grid and standard remapping

advection. Simulation results match the predicted values of a novel analytical solution for one-grid-cell-wide channels.

## 1 Introduction

CICE (Hunke et al., 2023) is a dynamic and thermodynamic sea ice model used for a variety of applications such as climate

modeling (e.g., DeRepentigny et al., 2020), sub-seasonal sea ice forecasting (e.g., Barton et al., 2021) and short-term sea ice forecasting (e.g., Smith et al., 2021). Since 2017, the model has been developed by the CICE Consortium, a group of institu-





tions from the USA, Canada, Denmark, and Poland.

Earlier versions of CICE used the Arakawa B-grid (Arakawa and Lamb, 1977, i.e., the horizontal velocity components $u$
and $v$ are co-located at cell corners) for the spatial discretization. Recently, many users have requested a C-grid capability,
in which the $u$ component is defined on east and west cell edges and the $v$ component on north and south edges. The C-grid
offers several advantages over the B-grid. First, it allows straightforward coupling with C-grid ocean models (e.g., NEMO,
HYCOM, Madec, 2008; Metzger et al., 2014) and atmospheric models (e.g., GEM, McTaggart-Cowan et al., 2019). Second,
it can represent transport along channels that are only one grid cell wide. Finally, the C-grid discretization better represents
inertial–plastic compressive waves (Bouillon et al., 2009).

For these reasons, members of the CICE Consortium decided to implement a finite-difference C-grid capability in CICE,
presented here. Specifically, we describe the C-grid discretization of the momentum equation and the formulation of boundary
conditions, along with the modifications required to use the incremental remapping transport scheme (Lipscomb and Hunke,
2004) with the new discretization.

In our first implementation of remapping for the C-grid, the C-grid velocities on cell edges were interpolated to the cor-
ners, and remapping was then applied the same way as for the B-grid. This approach, however, does not simulate transport in
one-grid-cell-wide channels. Also, some idealized tests show the presence of a checkerboard pattern in fields such as sea ice
concentration.

A modal analysis of the perturbed set of (simplified) equations shows that these spurious modes are due to the interpolation
of C-grid velocities to the corners, i.e., spatial averaging. Here, we demonstrate that the checkerboard pattern can be eliminated
by adjusting departure regions and edge fluxes using the native C-grid velocities. We refer to this method as the edge flux
adjustment (EFA) method.

The EFA method ensures that the divergence implied by the remapping scheme is consistent with the divergence computed
by the dynamical solver for the momentum equation. Interestingly, this consistency for the divergence leads to another crucial
advantage; the flux corrections allows transport in one-grid-cell-wide channels. During our investigation of transport in these
narrow channels, we realized that, given some assumptions, analytical solutions exist for this simple configuration. These ana-
lytical solutions are useful for verifying the C-grid implementation of rheology and velocities along the channel.

The main contributions of this work are (1) a detailed description and validation of the CICE C-grid spatial discretization,
including the formulation of boundary conditions, (2) the derivation of analytical solutions for one-grid-cell-wide channels, (3)
the description of a checkerboard pattern associated with the interpolation of C-grid velocities when using the standard remap-
ping scheme, (4) a mathematical analysis of the spurious pattern, (5) a demonstration that edge flux adjustment eliminates the





checkerboard pattern and allows transport in one-grid-cell-wide channels, and (6) modifications to the remapping algorithm to improve its robustness.

This article is structured as follows. Section 2 introduces the momentum and stress equations (2.1) and the equations for transport (2.2). Section 3 briefly introduces the C-grid spatial and temporal discretizations; more details can be found in Appendix A. Section 4 describes our initial implementation of incremental remapping for the C-grid along with its weaknesses. These weaknesses are corrected with the edge flux adjustment method, explained in Sect. 5. Section 6 gives an overview of the different tests used to validate the new C-grid discretization. Concluding remarks and future work are given in Sect. 7. Appendices B presents a modal analysis of the remapping checkerboard pattern. Appendix C describes some modifications to improve the robustness of the remapping method. Finally, Appendix D introduces a novel analytical solution for one-grid-cell-wide channels.

## 2 Model equations

### 2.1 Momentum equation and rheology

The 2D sea ice momentum equation is given by

$$m\frac{D\mathbf{u}}{Dt} = -\mathbf{k} \times mf\mathbf{u} + \boldsymbol{\tau}_a + \boldsymbol{\tau}_w + \boldsymbol{\tau}_b + \nabla \cdot \boldsymbol{\sigma} - mg_e\nabla H_0, \tag{1}$$

where $m$ is the combined mass per m$^2$ of sea ice and snow, $\mathbf{u} = u\mathbf{i} + v\mathbf{j}$ is the horizontal velocity vector with components $u$ and $v$, $\mathbf{i}$, $\mathbf{j}$ and $\mathbf{k}$ are unit vectors respectively aligned with the x,y and z axes of the coordinate system, $f$ is the Coriolis parameter, $\boldsymbol{\tau}_a$ is the air stress, $\boldsymbol{\tau}_w$ is the water (or ocean) stress, $\boldsymbol{\tau}_b$ is a seabed stress which represents the effect of grounded pressure ridges, $\nabla \cdot \boldsymbol{\sigma}$ is the rheology term with horizontal stress components $\sigma_{11} = \sigma_{xx}$, $\sigma_{22} = \sigma_{yy}$ and $\sigma_{12} = \sigma_{xy}$, $g_e$ is the Earth's gravitational acceleration, and $\nabla H_0$ is the sea surface tilt.

When using the B-grid discretization in CICE, there are different approaches for representing sea ice rheology and for solving the momentum equation: the viscous-plastic (VP, Hibler, 1979) rheology, which involves an implicit solution; the elastic-viscous-plastic framework (EVP, Hunke, 2001), which is based on the VP rheology but relies on an explicit method; the revised EVP approach (rEVP, Lemieux et al., 2012; Bouillon et al., 2013; Kimmritz et al., 2015), and the elastic-anisotropic-plastic model (EAP, Tsamados et al., 2013). In the current C-grid implementation, only the EVP and rEVP approaches are available. The EVP implementation is presented below.

Before describing the EVP equations for the internal stresses, we list a few modifications that were done recently to improve the flexibility of the VP (B-grid only) and (r)EVP (B-grid and C-grid) approaches. First, following König Beatty and Holland (2010), the yield curve can include tensile strength (see Lemieux et al. (2016) for details about the implementation in CICE).





Tensile strength improves the simulation of landfast ice in regions of deep water (Lemieux et al., 2016). Second, the stresses
are formulated in terms of viscosities, as introduced by Hibler (1979). Although only the elliptical yield curve is currently
available, the formulation with viscosities offers more flexibility for defining other yield curves (e.g., Zhang and Rothrock,
2005). Finally, the current implementation includes the plastic potential approach of Ringeisen et al. (2021). Due to the normal
flow rule, the standard VP rheology tends to simulate fracture angles that are too wide compared to observations (Ringeisen
et al., 2019; Hutter et al., 2022). This problem can be remedied with the use of the plastic potential, which defines post-fracture
deformations (or flow rule) independently from the yield curve. Following Ringeisen et al. (2021), the plastic potential is also
defined by an elliptical curve.

Given these latest innovations, the EVP equations for the internal stresses are given by

$$\frac{\partial \sigma_1}{\partial t} + \frac{\sigma_1}{2T_d} + \frac{p}{2T_d} = \frac{\zeta}{T_d} D_d, \tag{2}$$

$$\frac{\partial \sigma_2}{\partial t} + \frac{\sigma_2}{2T_d} = \frac{\eta}{T_d} D_t, \tag{3}$$

$$\frac{\partial \sigma_{12}}{\partial t} + \frac{\sigma_{12}}{2T_d} = \frac{\eta}{2T_d} D_s, \tag{4}$$

where $\sigma_1 = \sigma_{11} + \sigma_{22}$, $\sigma_2 = \sigma_{11} - \sigma_{22}$, $p$ is the replacement pressure (defined below), $\zeta$ and $\eta = e_G^{-2}\zeta$ are respectively the bulk
and shear viscosities, $e_G$ is the plastic potential's ellipse ratio of major to minor axes, and $T_d$ is a damping time scale for elastic
waves (Hunke, 2001). It is defined as $T_d = E_0\Delta t$ where $0 < E_0 < 1$ is a parameter and $\Delta t$ is the advective time step. The
strain rate terms $D_d = \frac{\partial u}{\partial x} + \frac{\partial v}{\partial y}$, $D_t = \frac{\partial u}{\partial x} - \frac{\partial v}{\partial y}$, and $D_s = \frac{\partial u}{\partial y} + \frac{\partial v}{\partial x}$ are the divergence, the horizontal tension, and the shearing
strain rate, respectively.

The bulk viscosity $\zeta$ is given by

$$\zeta = \frac{P(1+k_t)}{2\Delta}, \tag{5}$$

where $P$ is the ice strength, $k_t$ is a parameter between 0 and 1 that determines tensile strength (König Beatty and Holland, 2010)
and $\Delta$ is a deformation (i.e., strain rate) associated with the elliptical yield curve and expressed as $\Delta = \left[ D_d^2 + \frac{e_F^2}{e_G^4}(D_t^2 + D_s^2) \right]^{1/2}$
with $e_F$ the elliptical yield curve axis ratio. When $\Delta$ tends toward zero, $\zeta$ tends toward infinity. To prevent this singularity,
the denominator $\Delta$ in Eq. (5) is replaced by $\Delta^*$. There are two approaches in the code to define $\Delta^*$. By default, the capping
approach of Hibler (1979) is used. In this case, $\Delta^* = \max(\Delta, \Delta_{min})$ where $\Delta_{min}$ is a small deformation. A second approach
with $\Delta^* = (\Delta + \Delta_{min})$ allows a smoother formulation (Kreyscher et al., 2000). Finally, the replacement pressure $p$ ensures
that stresses are zero in the absence of deformations:





$$p = \frac{P(1-k_t)\Delta}{\Delta^*}. \tag{6}$$

The ice strength can be calculated with the approach of Hibler (1979) (referred to as H79) or with the Rothrock (1975) parameterization (referred to as R75). With H79, $P$ is given by

115 $$P = P^*\bar{h}e^{-C^*(1-a)}, \tag{7}$$

where $P^*$ and $C^*$ are parameters, $a$ is the sea ice concentration, and $\bar{h}$ is the mean thickness. Details about the more complicated R75 approach can be found in Rothrock (1975) and Lipscomb et al. (2007).

## 2.2 Transport equation

120 Roach et al. (2018) recently introduced in CICE a joint probability distribution $f(h,r)$ of sea ice thickness $h$ and floe size $r$. For simplicity, the transport equation is introduced here by only considering $g(h)$. Further information about horizontal transport in CICE can be found in Lipscomb and Hunke (2004).

The evolution of the ice thickness distribution $g(h)$ is given by

125 $$\frac{\partial g}{\partial t} = -\nabla \cdot (g\boldsymbol{u}) - \frac{\partial(gf_t)}{\partial h} + \psi, \tag{8}$$

where $f_t$ is the rate of thermodynamic ice growth and $\psi$ is a mechanical redistribution term. CICE solves this equation using an operator-splitting approach; the change of $g(h)$ from one time level to the next is computed in three steps where only one term on the right hand side is nonzero in each step. The last two terms in Eq. (8) are handled by the column physics model in CICE, called Icepack. Here, we only consider the change of $g(h)$ due to the horizontal transport term $-\nabla \cdot (g\boldsymbol{u})$.

130

Discretizing in terms of partial ice concentrations $a_n$, the transport equation for thickness category $n$ is given by

$$\frac{\partial a_n}{\partial t} + \nabla \cdot (a_n\boldsymbol{u}) = 0, \tag{9}$$

where $a_n$ is the ice concentration for category $n$. Snow volume, ice volume, snow enthalpy, and ice enthalpy also need to be transported, using equations of the form

135 $$\frac{\partial(a_nh_n)}{\partial t} + \nabla \cdot (a_nh_n\boldsymbol{u}) = 0, \tag{10}$$

$$\frac{\partial(a_nh_nq_n)}{\partial t} + \nabla \cdot (a_nh_nq_n\boldsymbol{u}) = 0, \tag{11}$$





where $h_n$ is the snow or ice thickness for category $n$ and $q_n$ is the enthalpy. The incremental remapping scheme solves Eqs. (9)–(11) in a unified way. Given the velocity field $\boldsymbol{u}$, departure regions are computed for each grid cell. Then the quantities $a_n$, $a_n h_n$, and $a_n h_n q_n$ are integrated over each departure region, so that volume and internal energy can be transferred conservatively between cells.

## 3 Spatial and temporal discretizations of the momentum equation

The spatial discretization of the new C-grid implementation is based on finite differences using curvilinear coordinates on a fixed Eulerian mesh. It differs from the B-grid implementation in the way the stresses, strain rates, and other terms in Eqs. (2)–(4) are discretized. Our implementation mostly follows the work of Bouillon et al. (2009), Bouillon et al. (2013) and Kimmritz et al. (2016). In this section we give a brief overview of the C-grid spatial and temporal discretizations of the momentum equation. Appendix A describes in detail the C-grid spatial discretization of the air stress (A1), the seabed stress (A2) and the rheology term as well as the time-stepping of the internal stresses (A3); A4 presents the time-stepping of the momentum equation. For both B- and C-grid implementations, the momentum equation is advanced in time first (with the ice thickness distribution held fixed), followed by the transport equation using the newly calculated sea ice velocity field.

Figure 1 shows where variables are defined for the C-grid discretization. Scalar variables such as ice thickness and ice concentration are defined at the tracer point $T$. Unlike the B-grid, where both velocity components are co-located at the corners (the $U$ points), the C-grid $u$ component is at the midpoint of the east ($E$) edge, while the $v$ component is at the midpoint of the north ($N$) edge. In the derivations below, a variable such as $u_E$ refers to the $u$ component of velocity evaluated on the east edge (Fig. 1), and similarly for variables with subscript $N$, $U$, or $T$, which are respectively evaluated at the north edge, the northeast corner, or the tracer point. The land-ocean mask is originally defined at the $T$ point and referred to as $M_T$, with $M_T = 1$ for ocean cells and $M_T = 0$ for land cells. Other masks ($M_U$, $M_E$, and $M_N$) are defined based on $M_T$. For example, $M_U = 1$ only if the four surrounding cells are ocean cells in the $M_T$ mask.

The most complex part of the C-grid implementation is the spatial discretization of the rheology term (A3). This involves the calculations of strain rates at the $U$ (A3.1) and $T$ (A3.2) points, of $\zeta$ and the replacement pressure at the $T$ points (A3.3), and of $\eta$ at the $U$ points (A3.5). No-slip/no-outflow boundary conditions are applied at the land-ocean boundaries using ghost velocities (see A3.1 for details). Following Bouillon et al. (2013), $D_s^2$ at a $T$ point is obtained from a spatial average of $D_s^2$ from the four neighboring $U$ points. This is done to enhance numerical stability. Following Kimmritz et al. (2016), the code includes two methods for calculating $\eta$ at the $U$ points; the default method averages $\eta_T$ from the neighboring ocean cells while the other approach is based on an averaging of the ice strength. Sections A3.4 and A3.6 describe the time-stepping of the stresses at the $T$ and $U$ points. The calculation of the $x$ and $y$ components of $\nabla \cdot \boldsymbol{\sigma}$ at the $E$ points ($F_{1E}$) and the $N$ points ($F_{2N}$) requires $\sigma_{1T}$, $\sigma_{2T}$, and $\sigma_{12U}$. Also, $\sigma_{12T}$ is calculated in order to diagnose normalized internal stresses (Lemieux and Dupont, 2020) at the $T$ points.





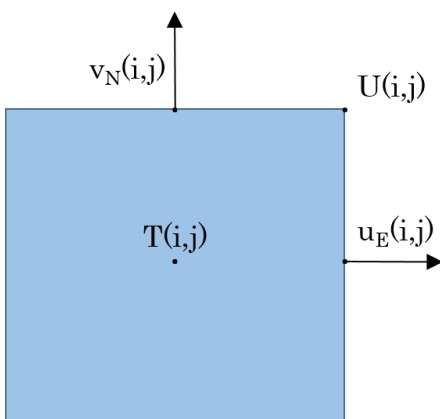

**Figure 1.** Schematic of a grid cell $(i, j)$ used for the spatial discretization. The indices $i$ and $j$ define the positions of variables respectively along the $x$ and $y$ axes. Scalars such as the ice concentration $a$ are defined at the $T$ point, while the C-grid velocity components $u_E(i, j)$ and $v_N(i, j)$ are respectively defined at the $E$ and $N$ points. The $U$ point, where both B-grid velocity components are located, is used for some C-grid variables such as the shear stress $\sigma_{12}$.

## 4  Initial approach for remapping using C-grid velocity components

Two transport schemes were already implemented in CICE: first-order upwind and incremental remapping. Upwind transport is naturally suited for a C-grid discretization, since edge velocities can be used directly to calculate fluxes on the four sides of a quadrilateral grid cell. Although upwind has some desirable characteristics (e.g., it is conservative, stable, computationally efficient, and sign-preserving), it is very diffusive. Sharp features such as the ice edge are quickly smeared out when simulating transport with an upwind scheme.

Like upwind, incremental remapping is conservative, stable, and sign-preserving. Although the geometric calculations (computing departure regions for each cell edge) are relatively expensive, the method scales well when there are many tracers, as is the case in CICE. Also, incremental remapping is much less diffusive than upwind and therefore preserves sharp features (Lipscomb and Hunke, 2004). The remapping algorithm for the C-grid implementation mostly follows that in Lipscomb and Hunke (2004). The differences are related to the calculation of departure regions, as described below.

### 4.1  Computation of departure regions

Departure regions are defined by approximating backward trajectories using corner velocities ($U$ points). As such, remapping is fundamentally a B-grid transport scheme. In our first implementation of remapping for the C-grid, C-grid velocities were interpolated to the corners, and remapping was used in the same way as for the B-grid. The velocity components $u_E$ and $v_N$





are interpolated to the $U$ points as

$$u_U(i,j) = M_U(i,j) \left[ \frac{u_E(i,j)A_E(i,j) + u_E(i,j+1)A_E(i,j+1)}{A_E(i,j) + A_E(i,j+1)} \right], \tag{12}$$

$$v_U(i,j) = M_U(i,j) \left[ \frac{v_N(i,j)A_N(i,j) + v_N(i+1,j)A_N(i+1,j)}{A_N(i,j) + A_N(i+1,j)} \right], \tag{13}$$

where $A_E$ and $A_N$ are grid cell areas evaluated at the $E$ and $N$ points, respectively, and the multiplication by $M_U(i,j)$ ensures

that the no-slip/no-outflow boundary conditions (BCs) are respected.

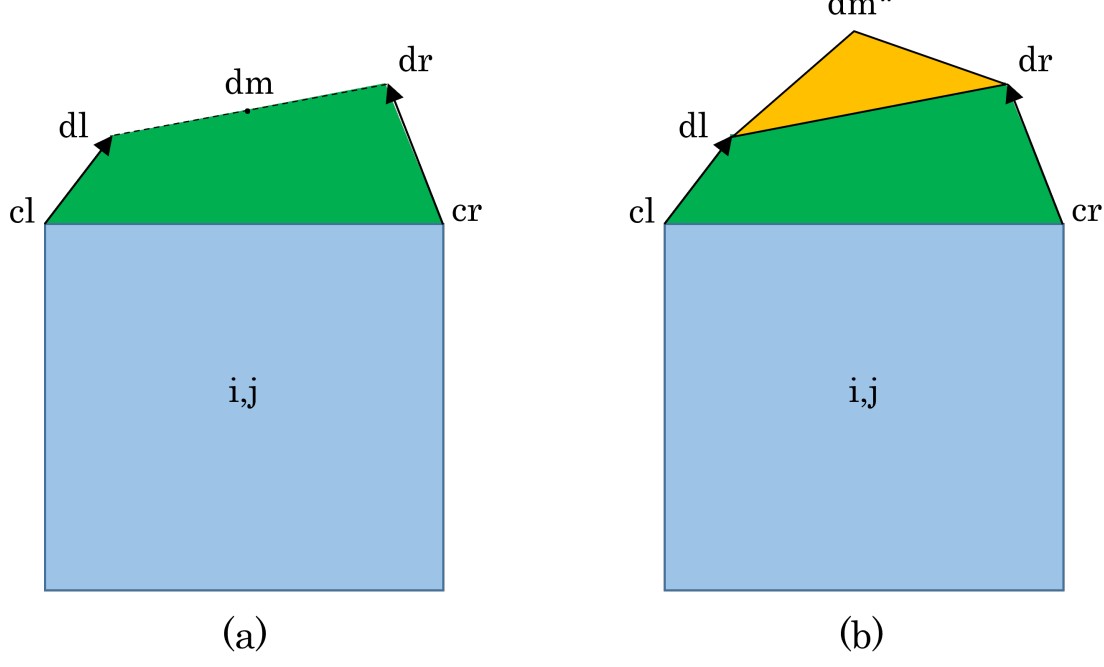

**Figure 2.** Schematic of departure regions (in green) on the north edge of grid cell $i,j$ (in blue) with the standard remapping (a) and the edge flux adjustment method (b). The departure regions are defined by the left corner point ($cl$), the right corner point ($cr$), the left departure point ($dl$), and the right departure point ($dr$). With the edge flux adjustment method, an additional triangle (in orange) is created by shifting the middle departure point $dm$ based on the edge flux associated with $v_N(i,j)$.

To improve the accuracy of the estimated departure regions, midpoints of the backward trajectories are computed first. Then, velocity components are bilinearly interpolated to these midpoints. Finally, these interpolated velocities are used to calculate the departure points defining the departure regions (Lipscomb and Hunke, 2004).






Panel (a) in Figure 2 shows an example of a departure region on the North edge of cell $(i, j)$. The departure region is a quadrilateral defined by the left ($cl$) and right ($cr$) corner points and the left ($dl$) and right ($dr$) departure points.

## 4.2 Weaknesses of the initial approach for remapping using C-grid velocity components

We identified two notable weaknesses with this initial C-grid discretization and remapping implementation. First, a C-grid discretization offers the possibility of representing transport in one-grid-cell-wide channels, but our initial implementation did not do so. Because of the no-slip/no-outflow BCs, the $U$ velocities are zero on both sides of such a channel, in which case the departure regions have zero area. Second, we found a numerical problem: in some idealized tests, we observed a checkerboard pattern in fields such as sea ice concentration. Panel (a) in Fig. 3 shows an example of this pattern, which indicates the presence

of spurious mathematical mode(s). This numerical noise is not present when using the upwind scheme (not shown).

Appendix B presents a modal analysis of a simplified set of perturbed equations (momentum and transport). We show that a stationary wave explains the formation of the checkerboard pattern. This stationary wave is a consequence of the spatial averaging used to obtain $u_U$ and $v_U$ (Eqs. (12) and (13)) for the remapping scheme.


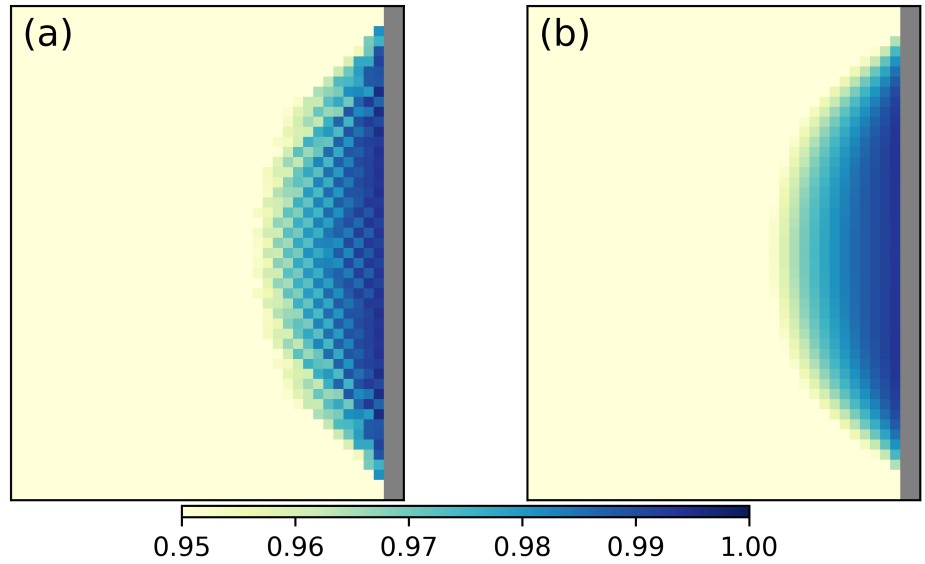

**Figure 3.** Sea ice concentration field after 15 days for a C-grid simulation with standard remapping (a) or with remapping and the edge flux adjustment method (b). The domain has 80×80 grid cells with $\Delta x = \Delta y = 16$ km. The simulation is initialized with a block of ice with $a = 0.8$ and $\bar{h} = 0.8$ m. The wind blows from the west with a magnitude of 5 m s$^{-1}$, the Coriolis parameter $f = 0$, and the ocean is at rest. The displayed range is 0.95 to 1 in order to better visualize the checkerboard pattern. This experiment is referred to as exp1. Some information (i.e., physical and numerical parameters) about this experiment is given in Table 1.





## 5 The edge flux adjustment method

### 5.1 Description of the method

Fortunately, the remapping transport scheme in CICE already included an option that eliminates the checkerboard pattern, which we refer to as the edge flux adjustment (EFA) method. With this method, the remapping scheme still calculates depar-
ture regions using the $U$ point velocities. However, the C-grid velocity components are then used to adjust the edge fluxes (or departure regions). Following the example in Fig. 2, the departure region is adjusted so that the total flux area $A_{tot}$ is equal to $v_N \Delta x_N \Delta t$, where $\Delta t$ is the advective time step and $v_N < 0$ in the example shown in Fig. 2. This is done by shifting the midpoint $dm$ of the line segment connecting the departure points $dl$ and $dr$. Geometrically speaking, this creates an additional departure triangle (the orange triangle in Fig. 2b).

The code computes the coordinates of the shifted $dm$ in a nondimensional coordinate system. In this coordinate system, the points $cl$ and $cr$ have coordinates $(-0.5, 0)$ and $(0.5, 0)$. The nondimensional flux area, denoted as $\tilde{A}_{tot}$, is equal to $A_{tot}/A_N$, where $A_N$ is the grid cell area evaluated at the $N$ point. Given nondimensional coordinates $(x_{dl}, y_{dl})$ and $(x_{dr}, y_{dr})$ for the departure points, the nondimensional coordinates $(x_{dm}^*, y_{dm}^*)$ for the shifted departure midpoint are obtained as

$$x_{dm}^* = x_{dm} + \alpha(y_{dr} - y_{dl}), \tag{14}$$

$$y_{dm}^* = y_{dm} - \alpha(x_{dr} - x_{dl}), \tag{15}$$

where $x_{dm} = (x_{dl} + x_{dr})/2$ and $y_{dm} = (y_{dl} + y_{dr})/2$ are the initial departure midpoint coordinates, and $\alpha$ is given by

$$\alpha = \frac{2\tilde{A}_{tot} + (x_{dr} - x_{cl})y_{dl} + (x_{cr} - x_{dl})y_{dr}}{(x_{dr} - x_{dl})^2 + (y_{dr} - y_{dl})^2}. \tag{16}$$

The adjusted area flux can finally be computed using the points $cl = (x_{cl}, y_{cl})$, $dl = (x_{dl}, y_{dl})$, $cr = (x_{cr}, y_{cr})$, $dr = (x_{dr}, y_{dr})$
and $dm^* = (x_{dm}^*, y_{dm}^*)$.

The initial departure region shown in Fig. 2a is confined to the central region (i.e., cell $(i, j)$ and/or cell $(i, j + 1)$). When part of the departure region (i.e., a triangle) is located for example in a corner cell (e.g., the northwestern cell $(i-1, j+1)$, not shown), the area of this corner triangle is subtracted from $A_{tot}$ before adjusting the central portion of the departure region (the
part lying in cell $(i, j + 1)$). In this case, we identify the point $(0, y_i)$ where the segment joining $dl$ and $dr$ intersects the left edge of cell $(i, j + 1)$. The departure point $dl$ is reset to $(0, y_i)$ before finding the shifted midpoint $dm^*$. Thus, the new triangle with vertices $(dl, dr, dm^*)$ is always located within the central region.

The calculation of $(x_{dm}^*, y_{dm}^*)$ as described above is the most common case, with both initial departure points on the same
side of the edge (i.e. $y_{dr}y_{dl} \geq 0$). We give another less common example in which $y_{dr}$ and $y_{dl}$ have opposite signs. Then there are two departure triangles: one on the left with vertices $(cl, dl, ip)$ and one on the right with vertices $(cr, dr, ip)$, where $ip$ denotes the point $(x_i, 0)$ where the departure segment intersects the $x$ axis.





In the case shown (Fig. 4), the nondimensional coordinate $x_i$ of the intersection point is greater than zero. There is a similar case not discussed here with $x_i < 0$.


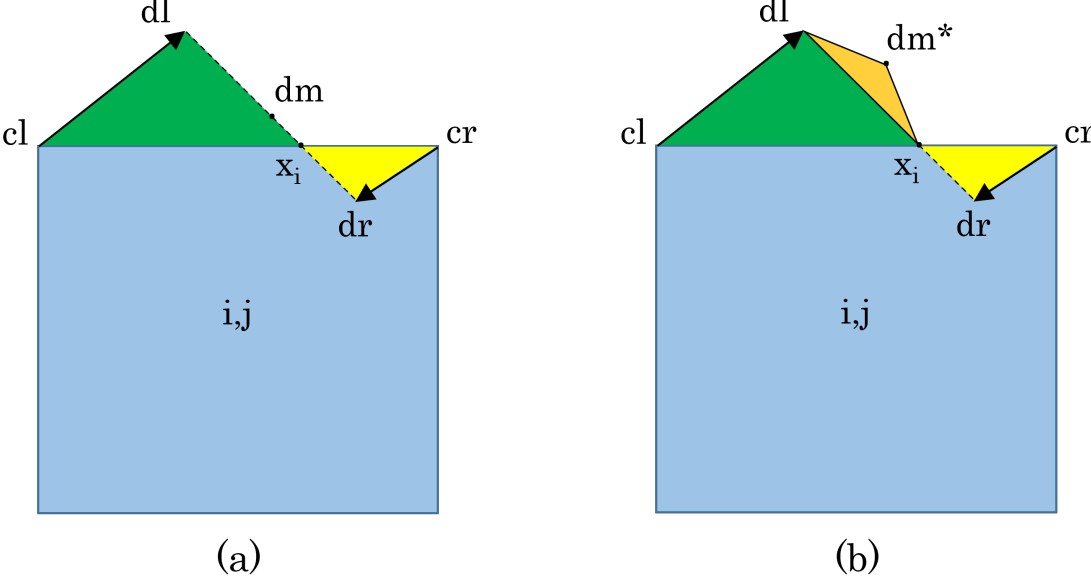

**Figure 4.** Schematic of departure regions on the north edge of grid cell $i, j$ (in blue) with the standard remapping (a) and the edge flux adjustment method (b). The departure region in (a) is defined by the green triangle (with vertices $cl$, $dl$, and $x_i$) and the yellow triangle (with vertices $cr$, $dr$, and $x_i$). With the EFA method in (b), the departure region has an additional triangle (orange, with vertices $x_i$, $dl$, and the shifted middle departure point $dm^*$). The orange triangle is calculated based on the edge flux associated with $v_N(i, j)$. The yellow triangle is referred to as the 'lone triangle'.

The strategy is to fix the right-hand triangle (the 'lone triangle') while modifying the left-hand triangle. Given the required total area flux $A_{tot} = v_N \Delta x_N \Delta t$ and the area $A_{r\triangle}$ of the right-hand triangle (in yellow), the EFA method first calculates the remaining area flux $A_l = A_{tot} - A_{r\triangle}$. The middle departure point is reset to $dm$ to $(\frac{x_{dl}+x_i}{2}, \frac{y_{dl}}{2})$, and then $dm$ is shifted so that the nondimensional area of the green and orange triangles is equal to $\tilde{A}_l = A_l/A_N$. The shifted middle departure point has

nondimensional coordinates

$$x^*_{dm} = x_{dm} - \alpha y_{dl}, \tag{17}$$

$$y^*_{dm} = y_{dm} - \alpha(x_i - x_{dl}), \tag{18}$$





where in this case $\alpha$ is given by

$$\alpha = \frac{2\tilde{A}_l + (x_i - x_{cl})y_{dl}}{(x_i - x_{dl})^2 + y_{dl}^2}. \tag{19}$$

When part of the departure region is a corner triangle lying outside cells $(i,j)$ and $(i,j+1)$, it is treated as described above for the most common case.

The EFA method ensures that the divergence (associated with edge fluxes) implied by the remapping is consistent with the divergence calculated by the dynamical solver (i.e., the EVP solver, see Eq. (A26)). Figure 3b shows that the EFA method prevents the formation of the checkerboard pattern. This pattern originates from an interaction between the solution of the

momentum equation and the standard remapping scheme. To support this conclusion, we conducted the following experiment: Velocity fields obtained with the C-grid discretization and the use of the EFA method in the remapping scheme were first stored. In a second simulation, both the dynamics (i.e., EVP) and EFA method are turned off, and the stored velocity fields are used for transport in the remapping scheme. In this case, no checkerboard pattern develops, and the concentration field is very similar to the one shown in Fig. 3b.


However, long-term C-grid simulations showed that the EFA method described above can sometimes fail on non-uniform grids. The remapping code in CICE includes many consistency checks to ensure that the solution is physically sound, for example that transport does not lead to negative area or mass.

With that initial implementation, the code failed on rare occasions with negative area and mass values close to land or the ice edge. These negative values were a result of approximations in the area of the departure region. Appendix C describes the changes that were required to improve the robustness of remapping.

## 5.2 Transport through one-grid-cell-wide channels

Interestingly, the EFA method also remedies the other weakness of our initial implementation: the absence of transport in one-grid-cell-wide channels. Although the initial departure regions have zero area (because the $U$ point velocities are zero along the channel edges), the departure regions are adjusted based on C-grid velocity components (e.g., the $v_N$ component would be non-zero for a north–south channel). In this case, the departure region is simply defined by $(x_{cl}, y_{cl})$, $(x_{cr}, y_{cr})$, and $(x_{dm}^*, y_{dm}^*)$ with $x_{dm}^* = 0$.


A minor code modification was nevertheless required. Before this modification, the edge fluxes were calculated only when at least one of the two departure points was displaced from its corner. Given non-displaced departure points for the north edge (i.e. $u_U(i-1,j) = v_U(i-1,j) = 0$ and $u_U(i,j) = v_U(i,j) = 0$), edge fluxes are now computed whenever $|v_N \Delta x_N \Delta t| > 0$. Similarly for the east edge (i.e. $u_U(i,j) = v_U(i,j) = 0$ and $u_U(i,j-1) = v_U(i,j-1) = 0$), edge fluxes are now computed





whenever $|u_E \Delta y_E \Delta t| > 0$.

To test this new capability, we implemented an idealized configuration with a long one-grid-cell-wide channel. The initial ice conditions are $a = 0.5$ and $\bar{h} = 1$ m over a length of five grid cells (80 km), with $a = 0$, $\bar{h} = 0$ m elsewhere. The wind blows from the west, and the fields are analysed after 30 days of simulation. As expected, there is no transport with the B-grid

discretization (not shown). Using the C-grid discretization, both the upwind method and remapping with the EFA method lead to transport in the channel (Fig. 5). As shown by Lipscomb and Hunke (2004) for more complex configurations, remapping is much less diffusive than upwind.

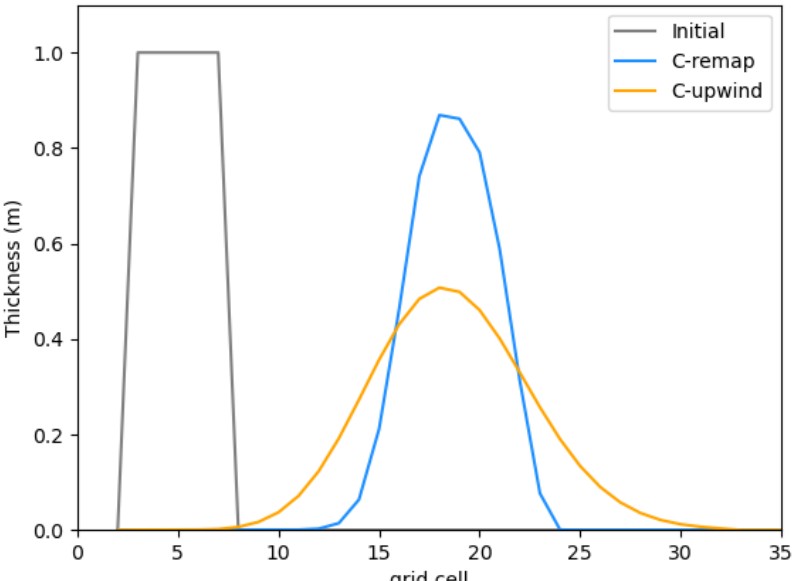

**Figure 5.** Simulated thickness after 30 days along a one-grid-cell-wide channel for the C-grid with remapping (blue) and upwind (orange). These simulations were initialized with $a = 0.5$ and $\bar{h} = 1$ m over a region five grid cells long (gray). The wind blows from the west at 5 m s$^{-1}$, $\Delta x = \Delta y = 16$ km, and the ocean is at rest. This experiment is referred to as exp2 in Table 1.

## 6 Validation of the C-grid implementation

We validated the new C-grid implementation with the EFA method using several approaches, including (1) symmetry tests, (2) thorough comparison of C-grid simulations with the default B-grid simulations, (3) comparison of C-grid simulations with analytical solutions, and (4) diagnostics of simulated stress states. This section gives an overview of the different tests. For





all the simulations presented here, we used version 6.5.0 of CICE, which includes our C-grid modifications. Unless otherwise specified, we used default values for most physical and numerical parameters. In the experiments described below, we mostly

modified the strength parameterization, $P^*$, $\Delta_{min}$, the number of EVP subcycles ($n_{evp}$) and $E_0$, in order to broaden the variety of tests (e.g., ice strength parameterization) and in some cases to improve the numerical convergence of the EVP method ($n_{evp}$ and $E_0$). See Table 1 for experiment details.

We conducted many idealized tests to verify the symmetry of simulated fields. Figure 6 shows an example. The thickness

fields after 14 days are perfectly symmetrical (bit-for-bit) for the four oblique (i.e., northeast, southeast, southwest and northwest) wind directions. Similar tests with the wind blowing from the west, east, north, and south also give symmetrical results, but with small differences (maximum difference is $4\times10^{-4}$ m). Changing the capping method to the smoother formulation (i.e., capping_method = 'sum') leads to bit-for-bit results.

We also evaluated more realistic simulations with B-grid runs as references. We compared C-grid runs on $1°$ and $3°$ global

grids to B-grid runs initialized and forced by the same fields. The JRA-55 reanalysis (Kobayashi et al., 2015) is used for the atmospheric forcing fields while the ocean forcing was derived from a Community Earth System Model (CESM) simulation (Kay et al., 2015). Panels a) and b), respectively, in Fig. 7 show the total simulated sea ice volume for the Northern and Southern Hemispheres for a $1°$ B-grid simulation with remapping transport (reference, orange), a $1°$ B-grid simulation with upwind

transport (blue), and a $1°$ C-grid simulation with remapping transport (dashed violet). Only the C-grid simulation uses the EFA method. Compared to the reference B-grid simulation, changing the grid discretization has a smaller impact on the total volume than changing the advection scheme. This is particularly evident in the Southern Hemisphere.

This conclusion is further supported by spatial maps of sea ice thickness. The monthly mean ice thickness in December

2009 for a $1°$ C-grid simulation with remapping is qualitatively correct, with the thickest ice found north of Greenland and in the Canadian Arctic Archipelago (Fig. 8). When compared to the reference simulation (B-grid with remapping), the ice is thinner in the regions of thick ice (Fig. 9a) although these differences are generally less pronounced than those for a B-grid with upwind compared to the reference (Fig. 9b). The same is true for the Southern Hemisphere (not shown).

The checkerboard pattern was not visible in our initial $1°$ and $3°$ C-grid simulations without EFA. Instead, the error mani-

fested as much thicker ice in convergent regions, due to a feedback between excessive ridging associated with the checkerboard pattern in velocities (and hence divergence and convergence), and ice growth in open water formed through the ridging process (not shown).

The most complex part of the C-grid implementation is the discretization of the rheology term. A crucial test is to verify

that the simulated internal stresses are inside (viscous) or on (plastic) the elliptical yield curve. To improve the numerical convergence of the EVP solver, the number of subcycling iterations $n_{evp}$ was increased from 240 (default) to 1200 and the





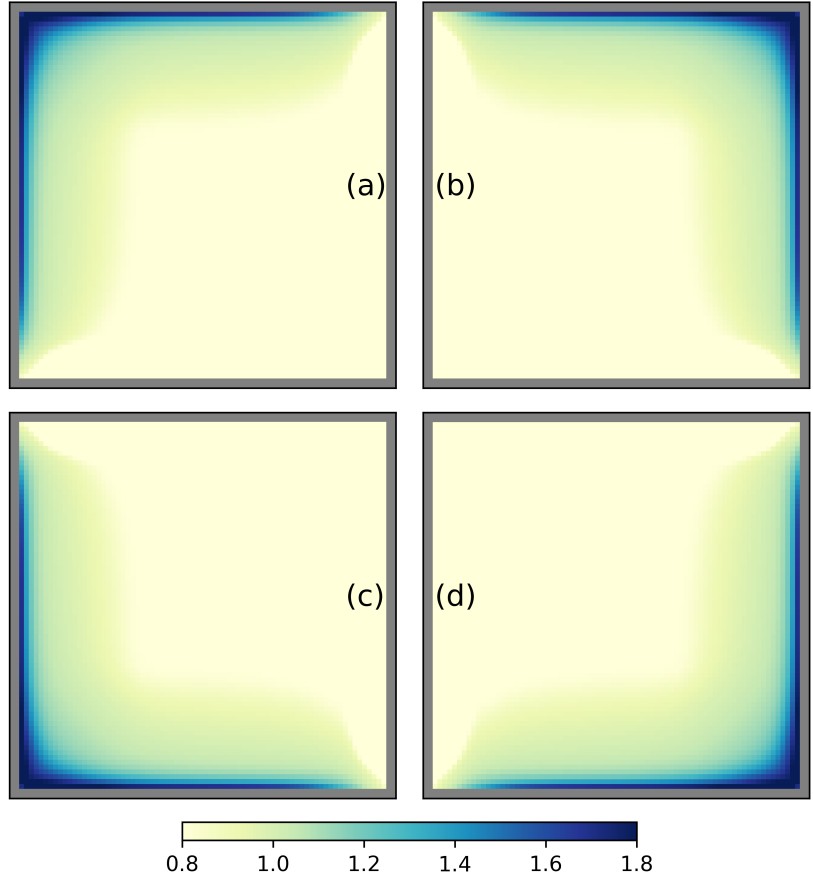

**Figure 6.** Simulated sea ice thickness after 14 days for winds blowing toward the northwest (a), northeast (b), southwest (c), and southeast (d). The domain has $80\times80$ grid cells with $\Delta x = \Delta y = 16$ km. The simulation is initialized with uniform ice conditions with $a = 0.8$ and $\bar{h} = 0.8$ m. The wind components have values of $\pm 5$ m s$^{-1}$. The ocean is at rest and $f = 0$. This experiment is referred to as exp3 in Table 1.

elastic damping parameter $E_0$ was reduced from 0.36 (default) to 0.09 (exp5 in Table 1). Following the approach of Lemieux and Dupont (2020), we plotted the states of stress, in stress-invariant coordinates, of a snapshot after five days of a 1° C-grid

simulation. Figure 10 confirms that the solution is viscous-plastic.

Analytical solutions are useful tools for verifying a numerical implementation. We derived novel analytical solutions for a one-grid-cell-wide channel with cyclic boundary conditions; see Appendix D for the derivation. As sea ice conditions are assumed constant in space and time, these solutions cannot be used to verify the simulated transport. Nevertheless, they pro-

vide steady-state analytical velocity values. Following the assumptions of this analytical solution with $a = 0.8$ and $\bar{h} = 0.8$ m for an east–west channel, the ice should be in the plastic (viscous) regime for winds larger (lower) than $u_{a*} = 3.176$ ms$^{-1}$.



Given $u_a = 4$ ms$^{-1}$ (i.e., $u_a > u_{a*}$), the steady-state analytical solution $u_E$ calculated independently using a Python code is 0.0409...ms$^{-1}$. The steady-state solution $u_E$ obtained with the new C-grid implementation matches the analytical solution up to the $12^{th}$ digit. Similarly, for $u_a = 1.5$ ms$^{-1}$ (i.e., $u_a < u_{a*}$), the steady-state analytical solution $u_E$ is much smaller and is equal to $7.135 \times 10^{-6}$ ms$^{-1}$. The new C-grid steady-state solution $u_E$ matches the analytical solution up to the $9^{th}$ digit (a difference less than $10^{-16}$ ms$^{-1}$). The same results are found for a north–south channel. This experiment is referred to as exp6 in Table 1.

| | strength | $P^*$ | $\Delta_{min}$ | $n_{evp}$ | $E_0$ |
|---|---|---|---|---|---|
| exp1 | H79 | 10 kNm$^{-2}$ | $2 \times 10^{-9}$ s$^{-1}$ | 1200 | 0.12 |
| exp2 | H79 | 27.5 kNm$^{-2}$ | $1 \times 10^{-11}$ s$^{-1}$ | 240 | 0.36 |
| exp3 | H79 | 10 kNm$^{-2}$ | $2 \times 10^{-9}$ s$^{-1}$ | 1200 | 0.12 |
| exp4 | R75 | - | $1 \times 10^{-11}$ s$^{-1}$ | 240 | 0.36 |
| exp5 | R75 | - | $2 \times 10^{-9}$ s$^{-1}$ | 1200 | 0.09 |
| exp6 | H79 | 27.5 kNm$^{-2}$ | $2 \times 10^{-9}$ s$^{-1}$ | 1200 | 0.12 |

**Table 1.** Table of some physical and numerical parameters used for the experiments.

## 7 Conclusions

We have designed and implemented a C-grid version of the CICE sea ice model. The C-grid spatial discretization is based on a finite-difference approach and follows the work of Bouillon et al. (2009, 2013) and Kimmritz et al. (2016). This article describes the finite-difference spatial discretization of the momentum equation, the implementation of no-slip/no-ouflow boundary conditions, and the use of the incremental remapping transport scheme with C-grid velocities.

Preliminary results from idealized experiments showed that the new C-grid discretization for the momentum equation and the use of the standard remapping transport scheme could produce checkerboard patterns in fields such as ice concentration. This numerical noise is not present when using the upwind transport scheme. A modal analysis of a simplified set of perturbed equations (i.e., momentum and transport with spatial discretization) shows that a stationary wave is responsible for the checkerboard pattern. This stationary wave results from the interpolation of C-grid velocity components to the $U$ points for use with remapping, which is fundamentally a B-grid scheme (Dukowicz and Baumgardner, 2000; Lipscomb and Hunke, 2004). This interpolation can be viewed as a spatial averaging. Many authors (e.g., Batteen and Han (1981)) have demonstrated that spatial averaging can lead to checkerboard patterns when solving the shallow-water equations, which are similar to our simplified set of equations.





To eliminate the checkerboard pattern, we modified the remapping scheme with an edge flux adjustment (EFA) method. This method uses C-grid velocity components at their natural locations to modify the departure regions calculated by the remapping, such that the implied divergence in the remapping is consistent with the divergence calculated by the dynamical solver. We also introduced some modifications to the calculation of the area of departure regions to increase the robustness of the EFA method on non-uniform grids.


     A C-grid discretization offers the possibility of representing transport in one-grid-cell-wide channels. Because of the no-slip/no-outflow boundary conditions, the $U$ point velocities at the channel edges are zero, and there is therefore no transport when using the standard remapping. The EFA method, however, allows transport in these channels, by creating departure regions with nonzero area based on the C-grid velocities. Moreover, remapping with the EFA method is much less diffusive than

upwind for idealized channel tests.

     For VP sea ice models, there are few existing analytical solutions due to the complexity of the rheology. We derived novel analytical solutions for one-grid-cell-wide channels and showed that with several simplifications (uniform ice conditions, constant wind, cyclic boundary conditions, and transport turned off), the sea ice velocity can be obtained analytically for both

plastic and viscous regimes. The steady-state values simulated by CICE match the analytical ones.

     We also verified that simulated internal stresses are consistent with the VP rheology (Hibler, 1979): the states of stress should lie either inside (viscous regime) or on the yield curve (plastic regime). Stresses outside the ellipse would be unrealistic. Plots of normalized stresses in stress-invariant coordinates (Lemieux and Dupont, 2020) confirm that the solution is consistent with

the VP rheology using an elliptical yield curve.

     We also conducted multiyear 1° global simulations to compare the C-grid solution (using the EFA method) to the reference B-grid solution (with standard remapping). We ran an additional B-grid simulation with upwind transport. Compared to the reference B-grid run, the C-grid discretization has a smaller impact on the total volume (and spatial differences) than changing

the advection scheme from remapping to upwind, especially in the Southern Hemisphere.

     Ongoing work within CICE Consortium modeling centers includes coupling this new CICE C-grid implementation to ocean models such as MOM6 and NEMO and to atmospheric models such as GEM.

**Appendix A: Spatial and temporal discretizations of the momentum equation**

The spatial discretization is presented below in the same order as in the code.





### A1 Air stress at the $E$ and $N$ points

For both B-grid and C-grid implementations, the air stress is calculated at the $T$ point and then interpolated to the required locations. For the C-grid, $\tau_{ax}$ at the $E$ point and $\tau_{ay}$ at the $N$ point are weighted averages of the values at the neighboring $T$ points and are given by

$$\tau_{axE}(i,j) = \frac{a_E(i,j)}{2A_E(i,j)} \left[ \tau_{axT}(i,j)A_T(i,j) + \tau_{axT}(i+1,j)A_T(i+1,j) \right], \tag{A1}$$

$$\tau_{ayN}(i,j) = \frac{a_N(i,j)}{2A_N(i,j)} \left[ \tau_{ayT}(i,j)A_T(i,j) + \tau_{ayT}(i,j+1)A_T(i,j+1) \right], \tag{A2}$$

where $A_E$, $A_N$, and $A_T$ are cell areas evaluated at the $E$, $N$ and $T$ points, and $a_E(i,j)$ and $a_N(i,j)$ are the ice concentration at the $E$ and $N$ points. These quantities are computed as

$$a_E(i,j) = \frac{a_T(i,j)A_T(i,j) + a_T(i+1,j)A_T(i+1,j)}{A_T(i,j) + A_T(i+1,j)}, \tag{A3}$$

$$a_N(i,j) = \frac{a_T(i,j)A_T(i,j) + a_T(i,j+1)A_T(i,j+1)}{A_T(i,j) + A_T(i,j+1)}. \tag{A4}$$

### A2 Seabed stress at the $E$ and $N$ points

The seabed stress components are $\tau_{bxE} = -C_{bE}u_E$ and $\tau_{byN} = -C_{bN}v_N$, where the $C_b$ coefficients are calculated as in Lemieux et al. (2016) or following the probabilistic approach of Dupont et al. (2022). For both approaches, $C_{bE}$ and $C_{bN}$ are written as

$$C_{bE}(i,j) = \frac{T_{bE}(i,j)}{\left( \sqrt{u_E^2(i,j) + v_E^2(i,j)} + u_0 \right)}, \tag{A5}$$

$$C_{bN}(i,j) = \frac{T_{bN}(i,j)}{\left( \sqrt{u_N^2(i,j) + v_N^2(i,j)} + u_0 \right)}, \tag{A6}$$

where $T_{bE}$ and $T_{bN}$ are factors that characterize the maximum possible seabed stress, and $u_0$ is a small velocity parameter that ensures a smooth transition between the static and dynamic regimes of the seabed stress. The velocity $v_E$ is obtained by interpolating $v_N$ to the $E$ point, and $u_N$ is found similarly by interpolating $u_E$ to the $N$ point. Using spatial weighted averages, $v_E$ and $u_N$ can be concisely written as





$$v_E(i,j) = \frac{1}{A_{Ntot}} \sum_{k=0}^{1} \sum_{l=-1}^{0} v_N(i+k,j+l) A_N(i+k,j+l) M_N(i+k,j+l), \tag{A7}$$

$$u_N(i,j) = \frac{1}{A_{Etot}} \sum_{k=-1}^{0} \sum_{l=0}^{1} u_E(i+k,j+l) A_E(i+k,j+l) M_E(i+k,j+l), \tag{A8}$$

where $A_{Ntot} = \sum_{k=0}^{1} \sum_{l=-1}^{0} A_N(i+k,j+l)$ and $A_{Etot} = \sum_{k=-1}^{0} \sum_{l=0}^{1} A_E(i+k,j+l)$. For example, to clarify the notation,

$A_{Ntot} = \sum_{k=0}^{1} \sum_{l=-1}^{0} A_N(i+k,j+l) = A_N(i,j-l) + A_N(i,j) + A_N(i+1,j-l) + A_N(i+1,j)$.

As opposed to the denominators in Eq. (A5), and (A6), the factors $T_{bE}$ and $T_{bN}$ do not vary during the EVP subcycling. They are therefore calculated before the subcycling. Following the approach of Lemieux et al. (2016), $T_{bE}$ and $T_{bN}$ are

$$T_{bE}(i,j) = k_2 \max[0, (h_E - h_{cE})] e^{-\alpha_b(1-a_E)}, \tag{A9}$$

$$T_{bN}(i,j) = k_2 \max[0, (h_N - h_{cN})] e^{-\alpha_b(1-a_N)}, \tag{A10}$$

where $h_E = \max[h_T(i,j), h_T(i+1,j)]$, $a_E = \max[a_T(i,j), a_T(i+1,j)]$, $h_{cE} = a_E h_{wE}/k1$, $h_{wE} = \min[h_w(i,j), h_w(i+1,j)]$, $h_N = \max[h_T(i,j), h_T(i,j+1)]$, $a_N = \max[a_T(i,j), a_T(i,j+1)]$, $h_{cN} = a_N h_{wN}/k1$ and $h_{wN} = \min[h_T(i,j), h_T(i,j+1)]$. $h_E$ ($h_N$), $h_{cE}$ ($h_{cN}$) and $h_{wE}$ ($h_{wN}$) are the mean ice thickness (or ice volume), the critical thickness, and the water depth at the $E$ ($N$) point, respectively.


When the seabed stress is computed based on the probabilistic approach, the calculation of $T_b$ factors is more complicated than with the method of Lemieux et al. (2016). Details can be found in Dupont et al. (2022). With the probabilistic approach, $T_b$ factors are first calculated at the $T$ point, and $T_{bE}$ and $T_{bN}$ are then given by

$$T_{bE}(i,j) = \max(T_{bT(i,j)}, T_{bT(i+1,j)}), \tag{A11}$$


$$T_{bN}(i,j) = \max(T_{bT(i,j)}, T_{bT(i,j+1)}). \tag{A12}$$

## A3    Discretization of rheology

As opposed to the variational method used for the B-grid (Hunke and Dukowicz, 1997, 2002), our C-grid spatial discretization is based on finite differences. With this approach, the discretization of $\nabla \cdot \boldsymbol{\sigma}$ requires the calculation of $\sigma_1$ and $\sigma_2$ at the $T$

points and of $\sigma_{12}$ at the $U$ points. The stresses are calculated in three steps: the computation of strain rates, the computation





of viscosities and replacement pressure, and finally the time-stepping of the stresses from subcycle $k$ to $k+1$. The subsections below explain how this is done for the $T$ and $U$ points. The sequence of computations follows that in the code.

The spatial discretization requires the computation of strain rates and components of the rheology term in curvilinear coordinates. The strain rates are given by

$$D_d = \dot{\epsilon}_{11} + \dot{\epsilon}_{22} = \frac{1}{h_1 h_2} \left[ \frac{\partial}{\partial \xi_1}(h_2 u) + \frac{\partial}{\partial \xi_2}(h_1 v) \right], \tag{A13}$$

$$D_t = \dot{\epsilon}_{11} - \dot{\epsilon}_{22} = \frac{h_2}{h_1} \frac{\partial}{\partial \xi_1}\left(\frac{u}{h_2}\right) - \frac{h_1}{h_2} \frac{\partial}{\partial \xi_2}\left(\frac{v}{h_1}\right), \tag{A14}$$

$$D_s = 2\dot{\epsilon}_{12} = \frac{h_1}{h_2} \frac{\partial}{\partial \xi_2}\left(\frac{u}{h_1}\right) + \frac{h_2}{h_1} \frac{\partial}{\partial \xi_1}\left(\frac{v}{h_2}\right), \tag{A15}$$

where $D_d$ is the divergence, $D_t$ is the tension, $D_s$ is the shear strain rate, $\xi_1$ and $\xi_2$ are nondimensional coordinates, and $h_1$ and $h_2$ are scale factors referred to as $\Delta x$ and $\Delta y$.

In curvilinear coordinates, the $x$ and $y$ components of the divergence of the stress tensor are respectively

$$F_1 = \frac{1}{2} \left[ \frac{1}{h_1} \frac{\partial \sigma_1}{\partial \xi_1} + \frac{1}{h_1 h_2^2} \frac{\partial(h_2^2 \sigma_2)}{\partial \xi_1} + \frac{2}{h_1^2 h_2} \frac{\partial(h_1^2 \sigma_{12})}{\partial \xi_2} \right], \tag{A16}$$

$$F_2 = \frac{1}{2} \left[ \frac{1}{h_2} \frac{\partial \sigma_1}{\partial \xi_2} - \frac{1}{h_1^2 h_2} \frac{\partial(h_1^2 \sigma_2)}{\partial \xi_2} + \frac{2}{h_1 h_2^2} \frac{\partial(h_2^2 \sigma_{12})}{\partial \xi_1} \right], \tag{A17}$$

which are Eqs. (20) and (21) in Hunke and Dukowicz (2002).

### A3.1 Strain rates at the $U$ point

To solve the momentum equation, shear stresses $\sigma_{12}$ are needed at the $U$ points including at land-ocean boundaries. This implies that strain rates and shear viscosities must be computed at these locations. In the code, strain rates at the $U$ points are first calculated. The reason for doing this is to follow what is suggested in Bouillon et al. (2013); to enhance numerical stability, $D_{sT}^2$ in $\Delta_T$ is a weighted average of the $D_{sU}^2$ around it.

As described later in Sect. A3.5, there are two methods for calculating $\eta$ at the $U$ points. Following Kimmritz et al. (2016), we refer to these methods as C1 and C2. The C1 method requires only $D_{sU}$, while the C2 method requires $D_{dU}$, $D_{tU}$, and





$D_{sU}$. C1 is the default method, but for completeness we explain here how $D_{dU}$, $D_{tU}$, and $D_{sU}$ are computed.


To ease the implementation of the boundary conditions (BCs) in the code, strain rates at the $U$ point are calculated differently than at the $T$ point. To do so, the derivatives in Eqs. (A13), (A14), and (A15) are expanded. First, for the divergence, Eq. (A13), we expand the derivatives and write

$$D_d = \frac{1}{h_1 h_2} \left[ h_2 \frac{\partial u}{\partial \xi_1} + u \frac{\partial h_2}{\partial \xi_1} + h_1 \frac{\partial v}{\partial \xi_2} + v \frac{\partial h_1}{\partial \xi_2} \right]. \tag{A18}$$

The discretized form of divergence at the $U$ point $(i,j)$ is therefore

$$D_{dU}(i,j) = \frac{1}{h_{1U}(i,j)h_{2U}(i,j)} [h_{2U}(i,j)(u_N^*(i+1,j) - u_N^*(i,j)) + u_U(i,j)(h_{2N}(i+1,j) - h_{2N}(i,j)) +$$
$$h_{1U}(i,j)(v_E^*(i,j+1) - v_E^*(i,j)) + v_U(i,j)(h_{1N}(i,j+1) - h_{1N}(i,j))], \tag{A19}$$

where $u_N^*(i+1,j)$, $u_N^*(i,j)$, $v_E^*(i,j+1)$, and $v_E^*(i,j)$ are modified versions of $u_N(i+1,j)$, $u_N(i,j)$, $v_E(i,j+1)$, and $v_E(i,j)$ to take into account the BCs. This is explained below.

Similarly, to calculate $D_t$ at the $U$ points we expand the derivatives in Eq. (A14) and write the tension as

$$D_t = \frac{1}{h_1 h_2} \left[ h_2 \frac{\partial u}{\partial \xi_1} - u \frac{\partial h_2}{\partial \xi_1} - h_1 \frac{\partial v}{\partial \xi_2} + v \frac{\partial h_1}{\partial \xi_2} \right]. \tag{A20}$$

The discretized form of this equation is

$$D_{tU}(i,j) = \frac{1}{h_{1U}(i,j)h_{2U}(i,j)} [h_{2U}(i,j)(u_N^*(i+1,j) - u_N^*(i,j)) - u_U(i,j)(h_{2N}(i+1,j) - h_{2N}(i,j)) -$$
$$h_{1U}(i,j)(v_E^*(i,j+1) - v_E^*(i,j)) + v_U(i,j)(h_{1N}(i,j+1) - h_{1N}(i,j))]. \tag{A21}$$

Finally, for the shear strain rate we write Eq. (A15) as

$$D_s = \frac{1}{h_1 h_2} \left[ h_1 \frac{\partial u}{\partial \xi_2} - u \frac{\partial h_1}{\partial \xi_2} + h_2 \frac{\partial v}{\partial \xi_1} - v \frac{\partial h_2}{\partial \xi_1} \right], \tag{A22}$$

with the discretized form

$$D_{sU}(i,j) = \frac{1}{h_{1U}(i,j)h_{2U}(i,j)} [h_{1U}(i,j)(u_E^*(i,j+1) - u_E^*(i,j)) - u_U(i,j)(h_{1E}(i,j+1) - h_{1E}(i,j)) +$$
$$h_{2U}(i,j)(v_N^*(i+1,j) - v_N^*(i,j)) + v_U(i,j)(h_{2N}(i+1,j) - h_{2N}(i,j))]. \tag{A23}$$

Away from the land-ocean boundaries, $u_E^*(i,j) = u_E(i,j)$, $u_E^*(i,j+1) = u_E(i,j+1)$, $v_N^*(i,j) = v_N(i,j)$, etc. However, at ocean-land boundaries, no-slip, no-outflow BCs are implemented by setting $u_U(i,j) = v_U(i,j) = 0$ and by using ghost values





for the other terms. As an example, consider $v_N(i,j)$ and $v_N(i+1,j)$ when the T cells at $(i+1,j)$ and $(i+1,j+1)$ are land cells. We need $v_N(i,j)$ and $v_N(i+1,j)$ to calculate the $\partial v/\partial \xi_1$ term in $D_{sU}$. As $v_N(i,j)$ is in the ocean, $v_N^*(i,j) = v_N(i,j)$. However, as $v_N(i+1,j)$ is on land, it is not defined and must be formulated using the BCs. We assume that $v_N$ varies linearly at the ocean-land boundary. We therefore write $v_N = mx + b$ where $m$ is the slope and $b$ is the value of $v_N$ at $x = 0$ which is defined at the ocean-land boundary. Using the no-outflow condition implies that $b = 0$. Given $h_{1N}(i,j)/2$ (the distance between

the ocean-land boundary and the $N(i,j)$ point) and $h_{1N}(i+1,j)/2$ (the distance between the ocean-land boundary and the $N(i+1,j)$ point), it is easy to show that

$$v_N^*(i+1,j) = -v_N(i,j)\frac{h_{1N}(i+1,j)}{h_{1N}(i,j)}, \tag{A24}$$

where in the case of a uniform Cartesian grid, $v_N^*(i+1,j)$ is simply $v_N(i,j)$ multiplied by $-1$. To take into account all the possible cases, the mask at the $N$ point ($M_N$) is used for the final formulation of $v_N^*(i+1,j)$:

$$v_N^*(i+1,j) = v_N(i+1,j)M_N(i+1,j) + [M_N(i,j) - M_N(i+1,j)]M_N(i,j)\frac{h_{1N}(i+1,j)}{h_{1N}(i,j)}v_N(i,j), \tag{A25}$$

which reduces to $v_N^*(i+1,j) = v_N(i+1,j)$ away from the ocean-land boundary (i.e., all four $T$ cells are ocean cells).

### A3.2 Strain rates at the $T$ point

Using Eq. (A13), a finite-difference approximation of the divergence at the $T$ point is given by

$$D_{dT}(i,j) = \frac{1}{h_{1T}(i,j)h_{2T}(i,j)}\left[h_{2E}(i,j)u_E(i,j) - h_{2E}(i-1,j)u_E(i-1,j) + h_{1N}(i,j)v_N(i,j) - h_{1N}(i,j-1)v_N(i,j-1)\right]$$

$$\tag{A26}$$

Similarly, using Eq. (A14), the tension at the $T$ point is given by

$$D_{tT}(i,j) = \frac{h_{2T}(i,j)}{h_{1T}(i,j)}\left[\frac{u_E(i,j)}{h_{2E}(i,j)} - \frac{u_E(i-1,j)}{h_{2E}(i-1,j)}\right] - \frac{h_{1T}(i,j)}{h_{2T}(i,j)}\left[\frac{v_N(i,j)}{h_{1N}(i,j)} - \frac{v_N(i,j-1)}{h_{1N}(i,j-1)}\right]. \tag{A27}$$

Following Bouillon et al. (2013), $D_{sT}^2$ is obtained as a weighted average of the neighboring $D_{sU}^2$:

$$D_{sT}^2(i,j) = \frac{1}{A_{Utot}}\sum_{k=-1}^{0}\sum_{l=-1}^{0}D_{sU}^2(i+k,j+l)A_U(i+k,j+l), \tag{A28}$$

where $A_U(i,j)$ is the cell area evaluated at the $U$ point and $A_{Utot} = \sum_{k=-1}^{0}\sum_{l=-1}^{0}A_U(i+k,j+l)$. At the $T$ point, the strain rate $\Delta_T$ for the viscosities is then calculated as

$$\Delta_T(i,j) = \left[D_{dT}^2(i,j) + \frac{e_F^2}{e_G^4}(D_{tT}^2(i,j) + D_{sT}^2(i,j))\right]^{1/2}. \tag{A29}$$





### A3.3 Viscosities and replacement pressure at the $T$ point

Given $\Delta_T(i,j)$ as calculated in Eq. (A29), $\zeta_T(i,j)$ with the capping approach of Hibler (1979) is obtained as

$$\zeta_T(i,j) = \frac{(1+k_t)P_T(i,j)}{\max(\Delta_T(i,j),\Delta_{min})}, \tag{A30}$$

where the ice strength $P_t$ is also evaluated at the $T$ point. Similarly, the replacement pressure at the $T$ point is

$$p_T(i,j) = \frac{(1-k_t)P_T(i,j)}{\max(\Delta_T(i,j),\Delta_{min})}\Delta_T(i,j). \tag{A31}$$

If $\zeta_T$ and $p_T$ are regularized with the smoother approach, as in Kreyscher et al. (2000), the denominator $\max(\Delta_T(i,j),\Delta_{min})$ in Eqs. (A30) and (A31) is replaced by $(\Delta_T(i,j)+\Delta_{min})$. The approach of Hibler (1979) can be used by setting capping_method = 'max' in the namelist, while the smoother formulation is used by setting capping_method = 'sum'.

### A3.4 Time-stepping of the stresses at the $T$ point

For our C-grid implementation, only $\sigma_1$ and $\sigma_2$ are required at the $T$ point for time-stepping the velocity components using the momentum equation. Nevertheless, $\sigma_{12}$ is also computed at the $T$ point in order to calculate normalized stresses (Lemieux and Dupont, 2020) as diagnostics. Following Eqs. (2)–(4), the stresses at the $T$ point are time-stepped from subcycle $k$ to $k+1$ as

$$\frac{\sigma_{1T}^{k+1}(i,j)-\sigma_{1T}^{k}(i,j)}{\Delta t_e} + \frac{\sigma_{1T}^{k+1}(i,j)}{2T_d} + \frac{p_T(i,j)}{2T_d} = \frac{\zeta_T(i,j)D_{dT}(i,j)}{T_d}, \tag{A32}$$

$$\frac{\sigma_{2T}^{k+1}(i,j)-\sigma_{2T}^{k}(i,j)}{\Delta t_e} + \frac{\sigma_{2T}^{k+1}(i,j)}{2T_d} = \frac{\eta_T(i,j)D_{tT}(i,j)}{T_d}, \tag{A33}$$

$$\frac{\sigma_{12T}^{k+1}(i,j)-\sigma_{12T}^{k}(i,j)}{\Delta t_e} + \frac{\sigma_{12T}^{k+1}(i,j)}{2T_d} = \frac{\eta_T(i,j)D_{sT}(i,j)}{2T_d}, \tag{A34}$$

where $\Delta t_e$ is the subcycling time step and $\eta_T(i,j) = e_G^{-2}\zeta_T(i,j)$.

It is straightforward to solve the equations above for $\sigma_{2T}^{k+1}(i,j)$, $\sigma_{1T}^{k+1}(i,j)$ and $\sigma_{12T}^{k+1}(i,j)$. Note that $\zeta_T$, $\eta_T$, $p_T$ and strain rates in the equations above are calculated with a velocity field at iteration $k$.

### A3.5 Viscosities at the $U$ point

With our C-grid implementation, only the shear viscosity $\eta$ is needed at the $U$ point. Two methods in the code can be used to calculate $\eta_U$. The default method (visc_method = 'avg_zeta' in the namelist) is a weighted spatial average of the values at the $T$ points. This is the C1 method of Kimmritz et al. (2016) and is the same method used in Bouillon et al. (2013). With the C1





method, $\eta_U$ is obtained from a weighted average of the $\eta_T$ values in ocean cells around the $U$ points. This can be concisely written as

$$\eta_U(i,j) = \frac{1}{A_{Ttot}} \sum_{k=0}^{1} \sum_{l=0}^{1} \zeta_T(i+k,j+l) A_T(i+k,j+l) M_T(i+k,j+l), \tag{A35}$$

where $A_{Ttot} = \sum_{k=0}^{1} \sum_{l=0}^{1} A_T(i+k,j+l) M_T(i+k,j+l)$, and $\eta_T(i,j)$ is simply $e_G^{-2}\zeta_T(i,j)$.

The second method (visc_method = 'avg_strength' in the namelist) relies on a weighted spatial average of the ice strength values at the surrounding ocean $T$ points. This is the C2 method of Kimmritz et al. (2016) and also the method used in Bouillon et al. (2009). The ice strength at the $U$ point is given by

$$P_U(i,j) = \frac{1}{A_{Ttot}} \sum_{k=0}^{1} \sum_{l=0}^{1} P_T(i+k,j+l) A_T(i+k,j+l) M_T(i+k,j+l), \tag{A36}$$

where $A_{Ttot}$ is the same as for Eq. (A35) above.

Given $\Delta_U(i,j) = \left[ D_{dU}^2(i,j) + e_F^2 e_G^{-4}(D_{tU}^2(i,j) + D_{sU}^2(i,j)) \right]^{1/2}$, the shear viscosity at the $U$ point with capping_method = 'max' is given by

$$\eta_U(i,j) = e_G^{-2} \frac{(1+k_t)P_U(i,j)}{\max(\Delta_U(i,j), \Delta_{min})}. \tag{A37}$$

With capping_method = 'sum', it is given by

$$\eta_U(i,j) = e_G^{-2} \frac{(1+k_t)P_U(i,j)}{(\Delta_U(i,j) + \Delta_{min})}. \tag{A38}$$

### A3.6   Time-stepping of the stresses at the $U$ point

Using $\eta_U$ and $D_{sU}$, the shear stress at the $U$ point is advanced in time from subcycle $k$ to subcycle $k+1$ according to

$$\frac{\sigma_{12U}^{k+1}(i,j) - \sigma_{12U}^{k}(i,j)}{\Delta t_e} + \frac{\sigma_{12U}^{k+1}(i,j)}{2T_d} = \frac{\eta_U(i,j)D_{sU}(i,j)}{2T_d}, \tag{A39}$$

which can easily be solved for $\sigma_{12U}^{k+1}(i,j)$. Note that $\eta_U$ and $D_{sU}$ in the equation above are calculated with a velocity field at iteration $k$.

### A3.7   Divergence of the stress tensor

Once the stresses at the $T$ and $U$ points have been advanced in time from $k$ to $k+1$, the components of the rheology term can
be calculated. Eqs. (A16) and (A17) introduced earlier can be rewritten as

$$F_1 = \frac{1}{h_1 h_2} \left[ \frac{h_2}{2} \frac{\partial \sigma_1}{\partial \xi_1} + \frac{1}{2h_2} \frac{\partial(h_2^2 \sigma_2)}{\partial \xi_1} + \frac{1}{h_1} \frac{\partial(h_1^2 \sigma_{12})}{\partial \xi_2} \right], \tag{A40}$$





$$F_2 = \frac{1}{h_1 h_2} \left[ \frac{h_1}{2} \frac{\partial \sigma_1}{\partial \xi_2} - \frac{1}{2h_1} \frac{\partial (h_1^2 \sigma_2)}{\partial \xi_2} + \frac{1}{h_2} \frac{\partial (h_2^2 \sigma_{12})}{\partial \xi_1} \right]. \tag{A41}$$

Using finite differences, the discretized formulation of $F_1$ at the $E$ point is

$$
\begin{aligned}
F_{1E}(i,j) = \frac{1}{h_{1E}(i,j)h_{2E}(i,j)} \big[ &\frac{h_{2E}(i,j)}{2} [\sigma_{1T}(i+1,j) - \sigma_{1T}(i,j)] + \\
&\frac{1}{2h_{2E}(i,j)} [h_{2T}^2(i+1,j)\sigma_{2T}(i+1,j) - h_{2T}^2(i,j)\sigma_{2T}(i,j)] + \\
&\frac{1}{h_{1E}(i,j)} [h_{1U}^2(i,j)\sigma_{12U}(i,j) - h_{1U}^2(i,j-1)\sigma_{12U}(i,j-1)] \big],
\end{aligned}
\tag{A42}
$$

while the discretized formulation of $F_2$ at the $N$ point is

$$
\begin{aligned}
F_{2N}(i,j) = \frac{1}{h_{1N}(i,j)h_{2N}(i,j)} \big[ &\frac{h_{1N}(i,j)}{2} [\sigma_{1T}(i,j+1) - \sigma_{1T}(i,j)] - \\
&\frac{1}{2h_{1N}(i,j)} [h_{1T}^2(i,j+1)\sigma_{2T}(i,j+1) - h_{1T}^2(i,j)\sigma_{2T}(i,j)] + \\
&\frac{1}{h_{2N}(i,j)} [h_{2U}^2(i,j)\sigma_{12U}(i,j) - h_{2U}^2(i-1,j)\sigma_{12U}(i-1,j)] \big].
\end{aligned}
\tag{A43}
$$

### A4   Time-stepping of the momentum equation

When using the B-grid discretization, the sea ice momentum equation in CICE can be solved either explicitly with the EVP (or revised EVP) approach or implicitly with a Picard solver (similar to the one described in Lemieux et al. (2008)). For now,

only the EVP and revised EVP approaches are implemented for the C-grid discretization.

As this subsection describes the time-stepping, the grid indices $(i,j)$ are omitted to simplify the description. Hence, $u_E(i,j)$ and $v_N(i,j)$ are here referred to as $u_E$ and $v_N$. Neglecting the advection of momentum and introducing the EVP time-stepping, the momentum equations for the $u_E$ and $v_N$ components are

$$\frac{m_E u_E^{k+1}}{\Delta t_e} = \frac{m_E u_E^k}{\Delta t_e} + m_E f v_E^k + \tau_{axE} + \tau_{wxE} + \tau_{bxE} + F_1 - m_E g \frac{\partial H_0}{\partial x}, \tag{A44}$$

$$\frac{m_N v_N^{k+1}}{\Delta t_e} = \frac{m_N v_N^k}{\Delta t_e} - m_N f u_N^k + \tau_{ayN} + \tau_{wyN} + \tau_{byN} + F_2 - m_N g \frac{\partial H_0}{\partial y}, \tag{A45}$$

where the interpolated quantities $v_E$ and $u_N$ are calculated using Eqs. (A7) and (A8). The terms $m_E$ and $m_N$ are

$$m_E = \frac{m_T(i,j)A_T(i,j) + m_T(i+1,j)A_T(i+1,j)}{A_T(i,j) + A_T(i+1,j)}, \tag{A46}$$

$$m_N = \frac{m_T(i,j)A_T(i,j) + m_T(i,j+1)A_T(i,j+1)}{A_T(i,j) + A_T(i,j+1)}. \tag{A47}$$





All the terms in Eq. (A44) are evaluated at the $E$ point, while all the terms in Eq. (A45) are evaluated at the $N$ point. The seabed stress components are $\tau_{bxE} = -C_{bE}u_E^{k+1}$ and $\tau_{byN} = -C_{bN}v_N^{k+1}$, where the $C_b$ coefficients are calculated as in Lemieux et al. (2016) or following the probabilistic approach of Dupont et al. (2022). Decomposing the water stress term, Eqs. (A44) and (A45) can be written as

$$\left(\frac{m_E}{\Delta t_e} + C_{wE}\cos\theta_w + C_{bE}\right)u_E^{k+1} = \left(mf + C_{wE}\sin\theta_w\right)v_E^k + c_x, \tag{A48}$$

$$\left(\frac{m_N}{\Delta t_e} + C_{wN}\cos\theta_w + C_{bN}\right)v_N^{k+1} = -\left(m_N f + C_{wN}\sin\theta_w\right)u_N^k + c_y, \tag{A49}$$

where

$$c_x = m_E u_E^k/\Delta t_e + \tau_{axE} + C_{wE}\left(u_{wE}\cos\theta_w - v_{wE}\sin\theta_w\right) + F_1 - m_E g\partial H_0/\partial x,$$
$$c_y = m_N v_N^k/\Delta t_e + \tau_{ayN} + C_{wN}\left(u_{wN}\sin\theta_w + v_{wN}\cos\theta_w\right) + F_2 - m_N g\partial H_0/\partial y,$$
$$C_{wE} = a_e\rho_w C_{dw}[(u_E^k - u_{wE})^2 + (v_E^k - v_{wE})^2],$$
$$C_{wN} = a_n\rho_w C_{dw}[(u_N^k - u_{wN})^2 + (v_N^k - v_{wN})^2]. \tag{A50}$$

In a coupled framework, for example, $u_{wE}$, $v_{wE}$, $u_{wN}$, and $v_{wN}$ could come from a C-grid ocean model.

As opposed to what is done for the B-grid, the Coriolis term and part of the water stress are explicit (i.e., at iteration $k$) because $u_E$ and $v_N$ are not co-located. Introducing $l_E = \frac{m_E}{\Delta t_e} + C_{wE}\cos\theta_w + C_{bE}$, $l_N = \frac{m_N}{\Delta t_e} + C_{wN}\cos\theta_w + C_{bN}$, $r_E = m_E f + C_{wE}\sin\theta_w$, and $r_N = m_N f + C_{wN}\sin\theta_w$, Eqs. (A48) and (A49) become

$$l_E u_E^{k+1} = r_E v_E^k + c_x, \tag{A51}$$

$$l_N v_N^{k+1} = -r_N u_N^k + c_y, \tag{A52}$$

which can be solved easily for $u_E^{k+1}$ and $v_N^{k+1}$.

The explicit approach for the off-diagonal C-grid terms (as described above) is the same as used by Kimmritz et al. (2016). Note that for the C-grid, the semi-implicit approach of Bouillon et al. (2009) could be used to solve for $u^{k+1}$ and $v^{k+1}$ (see their Eqs. 34 and 35).

## Appendix B: Modal analysis of the remapping checkerboard pattern

We conducted many numerical experiments to understand and simplify the conditions that lead to the checkerboard pattern. The goal of this simplification is to allow us to perform a modal analysis and identify the cause of this spurious node. From the





original experiment with results shown in Fig. 3a, we simplify the problem by forcing the $v$ component of velocity and the shear

viscosity $\eta$ to be zero. Having $\eta = 0$ is equivalent to setting $e_G$ to infinity. The ice strength is parameterized according to Hibler (1979). We also assume that the concentration is close to 1 and that the ice is in a single thickness category. In experiments for which the wind pushes the ice against a wall, the checkerboard starts to develop close to the wall. It is therefore reasonable to assume that the ice is in the plastic regime when this occurs. Considering the plastic regime and the absence of shear stress, the stress $\sigma = \sigma_{11}$ is expressed as

$$\sigma = \frac{P}{2\Delta}D_d - \frac{P}{2}, \tag{B1}$$

which becomes $\sigma = -P$ because $\Delta = |D_d|$ and $D_d < 1$ (convergence).

We write the momentum and transport equations as

$$\rho h \frac{\partial u}{\partial t} = \tau_a + \tau_w - P^* \frac{\partial h}{\partial x}, \tag{B2}$$

$$\frac{\partial h}{\partial t} + \frac{\partial (hu)}{\partial x} = 0, \tag{B3}$$

where $P = P^* h$ as we assume that the concentration is close to 1.

We linearize these equations around $h_0$ and $u_0$. That is, $h = h_0 + h'$ and $u = u_0 + u'$, where $h'$ and $u'$ are small perturbations. We also neglect the perturbations related to the water stress. This surface stress term does not affect the conclusion of the following analysis. Neglecting $h'u'$ terms, we have

$$\rho h_0 \frac{\partial u'}{\partial t} + \rho h' \frac{\partial u_0}{\partial t} + P^* \frac{\partial h'}{\partial x} = 0, \tag{B4}$$

$$\frac{\partial h'}{\partial t} + \frac{\partial (h_0 u')}{\partial x} + \frac{\partial (h' u_0)}{\partial x} = 0. \tag{B5}$$

Because the ice is compact and subject to a no-flow boundary condition, it is reasonable to assume that the base state $u_0$ is zero close to the wall. As $h_0$ is constant in space, we finally have

$$\rho h_0 \frac{\partial u'}{\partial t} + P^* \frac{\partial h'}{\partial x} = 0, \tag{B6}$$


$$\frac{\partial h'}{\partial t} + h_0 \frac{\partial u'}{\partial x} = 0, \tag{B7}$$

Equations (B6) and (B7) are similar to the one-dimensional shallow-water equations (with Coriolis set to zero). Many authors have studied these equations and described the checkerboard patterns that depend on the spatial discretization (Schoenstadt,





1980; Batteen and Han, 1981; Walters and Carey, 1983; Le Roux et al., 2005).


We assume solutions of the form $u' = \hat{u}e^{-i\omega t}$ and $h' = \hat{h}e^{-i\omega t}$, where $i$ is the unit imaginary number. Following Batteen and Han (1981), we adopt a semi-discrete approach; we analyze only the impact of the spatial discretization and do not consider the time discretization. We first obtain

$$-i\omega\rho h_0\hat{u} + P^*\frac{\partial\hat{h}}{\partial x} = 0, \tag{B8}$$


$$-i\omega\hat{h} + h_0\frac{\partial\hat{u}}{\partial x} = 0. \tag{B9}$$

We write $\hat{u} = \tilde{u}e^{i(kx+ly)}$ and $\hat{h} = \tilde{h}e^{i(kx+ly)}$, where $\tilde{u}$ and $\tilde{h}$ define the amplitudes, and we conduct the analysis for a uniform Cartesian grid with grid cells of size $\Delta x \times \Delta y$. The origin of our $x$ and $y$ coordinate system is at the $T$ point of a grid cell, that is, the $T$ point is at $(0,0)$ while the $E$ and $U$ points are respectively at $(\frac{\Delta x}{2}, 0)$ and $(\frac{\Delta x}{2}, \frac{\Delta y}{2})$. Evaluating Eq. (B8) at the $E$
point, we obtain

$$-i\omega\rho h_0\tilde{u}e^{\frac{ik\Delta x}{2}} + \frac{P^*\tilde{h}}{\Delta x}[e^{ik\Delta x} - 1] = 0, \tag{B10}$$

which can be rearranged as

$$\omega\rho h_0\tilde{u} - \frac{2P^*\tilde{h}}{\Delta x}\sin(\frac{k\Delta x}{2}) = 0. \tag{B11}$$

If the standard remapping (our initial implementation) is used, the departure regions are defined by trapezoids in our
simple 1D problem. The shape of these trapezoids depends on the $U$ point velocities, which are calculated from the average C-grid velocities as $u_U(i,j) = \frac{[u_E(i,j)+u_E(i,j+1)]}{2}$. The area of the trapezoid on the east edge is therefore proportional to $\frac{[u_E(i,j-1)+2u_E(i,j)+u_E(i,j+1)]}{4}$. In this case, considering the (perturbed) fluxes for both edges in our simple 1D problem, Eq. (B9) can be written as

$$-i\omega\tilde{h} + \frac{h_0\tilde{u}}{4\Delta x}[e^{i(\frac{1}{2}k\Delta x+l\Delta y)} + 2e^{i(\frac{k\Delta x}{2})} + e^{i(\frac{1}{2}k\Delta x-l\Delta y)} - e^{i(-\frac{1}{2}k\Delta x+l\Delta y)} - 2e^{i(\frac{-k\Delta x}{2})} - e^{i(-\frac{1}{2}k\Delta x-l\Delta y)}] = 0, \tag{B12}$$

which becomes

$$\omega\tilde{h} - \frac{h_0\tilde{u}}{\Delta x}\sin(\frac{k\Delta x}{2})(1 + \cos(l\Delta y)) = 0, \tag{B13}$$

Using Eq. (B11) to replace $h_0\tilde{u}$ in Eq. (B13), we obtain the dispersion relation

$$\omega^2 = \frac{2P^*}{\rho(\Delta x)^2}\sin^2(\frac{k\Delta x}{2})[1 + \cos(l\Delta y)]. \tag{B14}$$

Considering the smallest possible wavelength in the $y$ direction ($\lambda = 2\Delta y$), the wavenumber $l$ is then $l = \pi/\Delta y$. With that
value of $l$, we have $\omega = 0$ in Eq. (B14), which means that this wave does not propagate: it is a stationary wave, explaining





the presence of the checkerboard pattern. Note that the smallest wavelength in the other direction, $\lambda = 2\Delta x$, is not a problem because $\sin^2(\frac{k\Delta x}{2}) = \sin^2(\frac{\pi}{2}) = 1$.

On the other hand, if the EFA method is used, the fluxes are based on rectangles defined by the $u_E$ velocity components.
Given the fluxes on the west and east edges, Eq. (B9) can be written as

$$-i\omega\tilde{h} + \frac{h_0\tilde{u}}{\Delta x}[e^{\frac{ik\Delta x}{2}} - e^{\frac{-ik\Delta x}{2}}] = 0, \tag{B15}$$

which can be rearranged as

$$\omega\tilde{h} - \frac{2h_0\tilde{u}}{\Delta x}\sin(\frac{k\Delta x}{2}) = 0. \tag{B16}$$

Using Eq. (B11) to replace $h_0\tilde{u}$ in Eq. (B16), we find the dispersion relation

$$\omega^2 = \frac{4P^*}{\rho(\Delta x)^2}\sin^2(\frac{k\Delta x}{2}). \tag{B17}$$

Compared to Eq. (B14), the dispersion relation associated with the EFA method (Eq. B17) does not have the $[1+\cos(l\Delta y)]$ term. As for the $\sin^2(\frac{k\Delta x}{2})$ term in Eq. (B14), the smallest wavelength $\lambda = 2\Delta x$ does not create a stationary wave.

## Appendix C: Improved robustness of remapping

Long-term C-grid simulations showed that the initial implementation of the EFA method sometimes failed on non-uniform grids. These rare failures were due to negative area and mass values close to land or the ice edge.

These negative values were a result of approximations in the area of the departure region. As explained in Subsect. 5.1, the points defining the departure triangles and the shifted departure midpoints are calculated in nondimensional coordinates.
Once the triangles have been found, their areas are scaled to the true grid dimensions, with an area factor $A_f$ assigned to each triangle. This factor is simply an approximation of the grid cell area at a certain location. Triangle areas $A_\triangle$ are calculated as

$$A_\triangle = \frac{A_f}{2}[(x_2 - x_1)(y_3 - y_1) - (y_2 - y_1)(x_3 - x_1)], \tag{C1}$$

where $(x_1, y_1)$, $(x_2, y_2)$, and $(x_3, y_3)$ are the nondimensional coordinates of the three triangle vertices.

To enhance the robustness of the remapping, the new code modifies some of the area factors. We show two examples to summarize the problems and solutions. In the first example (Fig. C1), we assume that the ocean cell $(i, j)$ has no ice in category $n$ before the transport step. We examine the transport calculation for that category. We assume that cells $(i-1, j)$ and $(i, j+1)$ are land cells, while cells $(i+1, j)$ and $(i, j-1)$ are ocean cells. This means that cell $(i, j)$ can have fluxes only across its east





and south edges. We finally assume that cell $(i+1,j)$ has ice in category $n$. On the south edge (Fig. C1a), the shifted middle

departure point $(dm^*)$ is in the same cell $(i,j)$ as the initial middle departure point. This reflects the fact that $v_N(i,j-1)$ has

the same sign as $v_U(i,j-1)$ (i.e., $v_N(i,j-1) < 0$ and $v_U(i,j-1) < 0$). Less commonly, the departure region on the east

edge is as shown in Fig. C1b, with $u_E(i,j) > 0$, $u_U(i,j-1) < 0$, and $|u_E(i,j)| > |u_U(i,j-1)|$. In that case, the initial middle

departure point $dm$ (not shown) in cell $(i+1,j)$ is shifted to $dm^*$ in cell $(i,j)$.

On the east edge of cell $(i,j)$, the orange triangle represents an area flux out of the cell, while the yellow triangle is an

incoming flux. On the south edge, all three triangles represent outgoing fluxes. As $a_n = 0$ in cell $(i,j)$, the fluxes associated

with the orange and the dark blue triangle are zero. The only triangles that matter are the yellow triangles associated with the

south and the east edges. The triangle associated with the south edge has vertices $(cr, dr, y_i)$, while the one associated with the

east edge has vertices $(cr, dr, x_i)$. In nondimensional coordinates, the incoming area flux across the east edge is larger than the

outgoing flux across the south edge. This should not lead to a negative net flux for cell $(i,j)$. However, if the area factors $A_f$

are different for the two yellow triangles, the outgoing area can exceed the incoming area, leading to negative ice area in cell

$(i,j)$.

As described in Sect. 5.1, the EFA method uses the cell area evaluated at the midpoint of the edge to calculate the nondi-

mensional area of the departure region. Both yellow and orange triangles in Fig. C1b therefore have $A_f = A_E(i,j)$, where

$A_E(i,j)$ is the cell area evaluated at the $E$ point. For the south edge, the dark blue and orange triangles have $A_f = A_N(i,j-1)$

(Fig. C1a). Because the yellow triangle is not in the central region of the south edge, it is not part of the adjustment process.

In the previous version of the remapping, the area factor assigned to this triangle was $A_f = A_U(i,j-1)$. On highly deformed

grids, $A_U(i,j-1)$ can be larger than $A_E(i,j)$, resulting in negative fluxes.


To improve the robustness of the remapping for C-grid simulations, we modified the code by assigning the area factor

$A_E(i,j)$ to the yellow triangle associated with the south edge. Similar modifications are required for triangles referred to as

TL (top left), BL (bottom left), TR (top right), and BR (bottom right) in the code. These modifications apply with or without

the EFA method.


Differing area factors are less of an issue for the B-grid. Considering the same example, the departure region for the east

edge would be defined by the points $cr$, $dr$, and $cl$ (Fig. C1b). The area flux into cell $(i,j)$ would be much larger than the area

flux out of the cell (yellow triangle in Fig. C1a), and there would be no negative fluxes.

We omit the details here, but a similar problem can arise with lone triangles (e.g., the yellow triangle in Fig. 4). This triangle

is now assigned $A_f = A_N(i,j)$ instead of $A_U(i,j)$, as was done before. All the lone triangles now use $A_f$ at the center of the

edge that they border.





As a result of the code modifications described above, another change was required to prevent negative areas. For simplicity, we omit the EFA triangles in this explanation; see Fig. C2. Here, we assume that the ocean cells $(i,j)$, $(i-1,j)$, and $(i+1,j)$ do not have ice in category $n$. Moreover, there are ocean cells to the north with $a_n > 0$, while the southern boundary is a coastline (i.e., land cells). We look at the fluxes for category $n$.

In rare situations, the segment joining $dl$ and $dr$ crosses two edges to form two corner triangles on the north edge of cell $(i,j)$, as shown in Fig. C2a. Since $a_n = 0$ in this cell, the departure region inside this cell associated with the north edge does not contribute to the total flux. This region, defined by the points $cl$, $dl$, $y_i$, and $cr$, is shown in dark blue in Fig. C2a. Similarly, since $a_n = 0$ in cell $(i+1,j)$, the green triangle for the north edge (Fig. C2a) and the green triangle for the east edge (Fig. C2b) do not contribute to the total flux. The two triangles that matter are the yellow ones. The one for the north edge, defined by $(cr, dr, x_i)$, represents a flux out of cell (i,j), while the one for the east edge defined by $(cl, dl, y_i)$, corresponds to an incoming flux. In nondimensional coordinates, the incoming area flux is greater than the outgoing flux. But if the two triangles have different area factors as described above, the net flux can be negative.

With the code changes described above, the yellow triangle for the east edge uses $A_f = A_N(i+1,j)$. To ensure robustness (i.e., positive areas) with these changes, the code now uses $A_f = A_N(i+1,j)$ for the yellow triangle associated with the north edge (Fig. C2a). This triangle is referred to as TR1 in the code. The green triangle, known as BR1, uses the area factor $A_E(i,j)$. Similar modifications are required for the analogous TL1, BL2, and BR2 triangles.

## Appendix D: C-grid analytical solution for a one-grid-cell-wide channel

We consider a uniform Cartesian grid with an east-west oriented, one-grid-cell-wide channel, applying cyclic boundary conditions. The wind is constant and blows from the west. We further simplify the problem by assuming that the ocean is at rest, the sea surface tilt term is zero, and the Coriolis parameter is zero. The ice conditions are considered constant along the channel with $a$ the concentration and $\bar{h}$ the mean thickness. For this test, these fields do not change in time, since transport, redistribution, and thermodynamics are turned off. Finally, we do not consider the plastic potential and simply set $e_G = e_F = e$. With these simplifications, $v = 0$ and the u-momentum equation becomes

$$m\frac{\partial u}{\partial t} = \tau_a + \tau_w + \frac{\partial \sigma_{11}}{\partial x} + \frac{\partial \sigma_{12}}{\partial y}. \tag{D1}$$

Because of the cyclic boundary conditions, $\frac{\partial \sigma_{11}}{\partial x} = 0$. At steady-state, the $u$-momentum equation therefore becomes

$$\tau_a + \tau_w + \frac{\partial \sigma_{12}}{\partial y} = 0. \tag{D2}$$

Discretizing Eq. (D2) at the $E$-point, we obtain

$$\tau_{aE} + \tau_{wE} + \frac{\sigma_{12U}(i,j) - \sigma_{12U}(i,j-1)}{\Delta y} = 0. \tag{D3}$$





At steady-state, the shear stresses are given by

$$\sigma_{12} = 2\eta\dot{\epsilon}_{12}, \tag{D4}$$

where $\eta$ is the shear viscosity and $\dot{\epsilon}_{12}$ is the shear strain rate. Note that in CICE, $D_S = 2\dot{\epsilon}_{12} = \partial u/\partial y + \partial v/\partial x$. Given the ellipse parameter $e$, $\eta$ is expressed as

$$\eta = \frac{e^{-2}P}{2\triangle^*}, \tag{D5}$$

where $P$ is the ice strength and $\triangle^* = \max(\triangle, \triangle_{min})$. Because $\dot{\epsilon}_{11}$ and $\dot{\epsilon}_{22}$ are zero, $\triangle = e^{-1}|D_S|$.

With $\tau_a = a\rho_a C_{da} u_a^2$ and $\tau_w = -a\rho_w C_{dw} u_E^2(i,j)$, Eq. (D3) can be written as

$$a\rho_a C_{da} u_a^2 - a\rho_w C_{dw} u_E^2(i,j) + \frac{\eta_U(i,j)D_{sU}(i,j) - \eta_U(i,j-1)D_{sU}(i,j-1)}{\Delta y} = 0. \tag{D6}$$

With the no-slip boundary condition, we can approximate the shear strain rate as

$$D_{sU}(i,j) = \frac{0 - u_E(i,j)}{\Delta y/2}, \tag{D7}$$

$$D_{sU}(i,j-1) = \frac{u_E(i,j) - 0}{\Delta y/2}, \tag{D8}$$

which means that $D_{sU}(i,j) < 0$ and $D_{sU}(i,j-1) = -D_{sU}(i,j)$.

We want to solve Eq. (D6) for $u_E(i,j)$. For simplicity, we drop $(i,j)$, i.e. $u_E(i,j) = u_E$. With strong winds, the ice is in the plastic regime, that is $\triangle^* = \triangle = e^{-1}|D_s|$. We can write Eq. (D6) as

$$a\rho_a C_{da} u_a^2 - a\rho_w C_{dw} u_E^2 - \frac{P}{e\Delta y} = 0. \tag{D9}$$

The transition between the plastic and viscous regimes occurs for a wind velocity $u_a = u_{a*}$. At this transition, $\triangle = \triangle_{min}$, which leads to a sea ice velocity of $e\triangle_{min}\Delta y/2$. Replacing $u_E$ in Eq. (D9) by that expression gives

$$a\rho_a C_{da} u_{a*}^2 - a\rho_w C_{dw} \left[\frac{e\triangle_{min}\Delta y}{2}\right]^2 - \frac{P}{e\Delta y} = 0. \tag{D10}$$

Solving for $u_{a*}$ we get

$$u_{a*} = \left[\frac{\rho_w C_{dw}}{\rho_a C_{da}}\left(\frac{e\triangle_{min}\Delta y}{2}\right)^2 + \frac{P}{a\rho_a C_{da}e\Delta y}\right]^{1/2}. \tag{D11}$$



If $u_a > u_{a*}$ the ice is in the plastic regime and $u_E$ can be found by solving Eq. (D9):

$$u_E = \left[ \frac{\rho_a C_{da} u_a^2}{\rho_w C_{dw}} - \frac{P}{a \rho_w C_{dw} e \Delta y} \right]^{1/2}, \tag{D12}$$

where the first term is the free-drift velocity and the second term, which is due to the rheology, slows down the ice. In the plastic regime, the shear stresses $\sigma_{12U}(i,j)$ and $\sigma_{12U}(i,j-1)$ are respectively $-e^{-1}P/2$ and $e^{-1}P/2$.

On the other hand, if the wind is weak (i.e, $u_a < u_{a*}$), the ice is in the viscous regime. In this case $\triangle^* = \triangle_{min}$ and Eq. (D6) becomes

$$a \rho_a C_{da} u_a^2 - a \rho_w C_{dw} u_E^2 - \frac{2 P u_E}{e^2 \triangle_{min} \Delta y^2} = 0, \tag{D13}$$

which can be rewritten as

$$u_E^2 + \frac{2P}{a \rho_w C_{dw} e^2 \triangle_{min} \Delta y^2} u_E - \frac{\rho_a C_{da} u_a^2}{\rho_w C_{dw}} = 0. \tag{D14}$$

The solution of Eq. (D14) is thus

$$u_E = -\frac{P}{a \rho_w C_{dw} e^2 \triangle_{min} \Delta y^2} + \sqrt{\left( \frac{P}{a \rho_w C_{dw} e^2 \triangle_{min} \Delta y^2} \right)^2 + \frac{\rho_a C_{da} u_a^2}{\rho_w C_{dw}}}. \tag{D15}$$

*Code and data availability.* The CICE code is available on GitHub at https://github.com/CICE-Consortium/CICE. The simulations for this article were done with release 6.5.0 which can be obtained at https://github.com/CICE-Consortium/CICE/releases/tag/CICE6.5.0 and on Zenodo at https://zenodo.org/records/10056499. Release 6.5.0 includes Icepack 1.4.0. The atmospheric forcing fields (JRA55) and CESM oceanic forcing fields used for the global simulations can be found on Zenodo at https://zenodo.org/records/8118239 and https://zenodo.org/records/4660188.

*Author contributions.* JFL led the design and implementation of the C-grid discretization with contributions from AC, DB, TR, PB and EH. AC implemented the test cases with contributions from DB, EH, DH, JFL, PB and RA. MB and WHL designed and implemented the first version of the edge flux adjustment method, and JFL and WHL improved the robustness of the method. JFL, FD and PB investigated the remapping checkerboard pattern. JFL derived the analytical solution for the one-grid-cell-wide channel. JFL wrote the article with contributions from all the co-authors.

*Competing interests.* The authors declare that they have no conflict of interest.

*Acknowledgements.* We would like to thank Carolin Mehlmann, Martin Losch and Sergey Danilov for interesting discussions about sea ice model C-grid and CD-grid implementations. DAB and WHL were supported by the NSF National Center for Atmospheric Research,



which is a major facility sponsored by the National Science Foundation under Cooperative Agreement No. 1852977. TR was funded by the Danish State through the National Centre for Climate Research (NCKF). MB was supported by the Research Council of Norway project INES (270061). EH was supported by the U.S. Department of Energy Office of Biological and Environmental Research, Earth System Model Development program. AC was funded through a National Oceanic and Atmospheric Administration contract in support of the CICE Consortium.





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

**Figure 7.** Simulated sea ice volume in the Northern Hemisphere (a) and the Southern Hemisphere (b) as a function of time for the B-grid with remapping transport (orange), the B-grid with upwind transport (blue) and the C-grid with remapping transport (dashed violet). These are 5-year simulations on a 1° global grid initialized from a long simulation. This experiment is referred to as exp4 in Table 1.

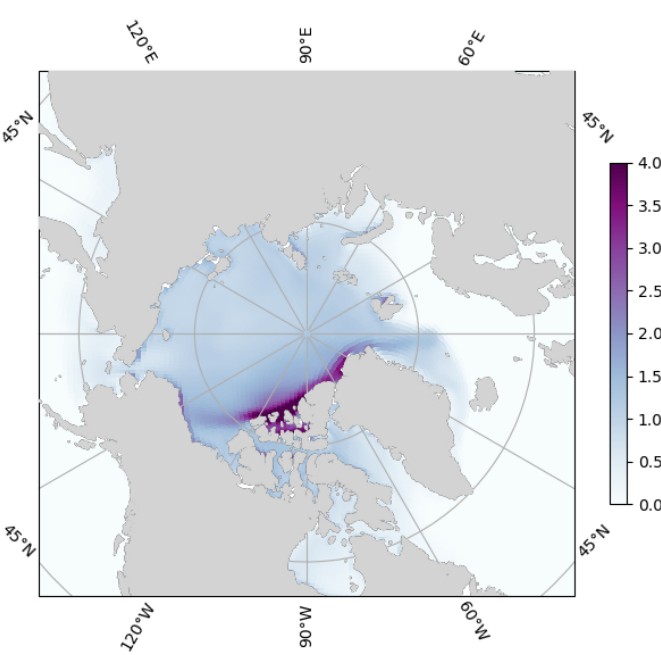

**Figure 8.** Monthly mean sea ice thickness (m) after five years (December 2009) for a 1° C-grid simulation with the remapping transport scheme. This experiment is referred to as exp4 in Table 1.



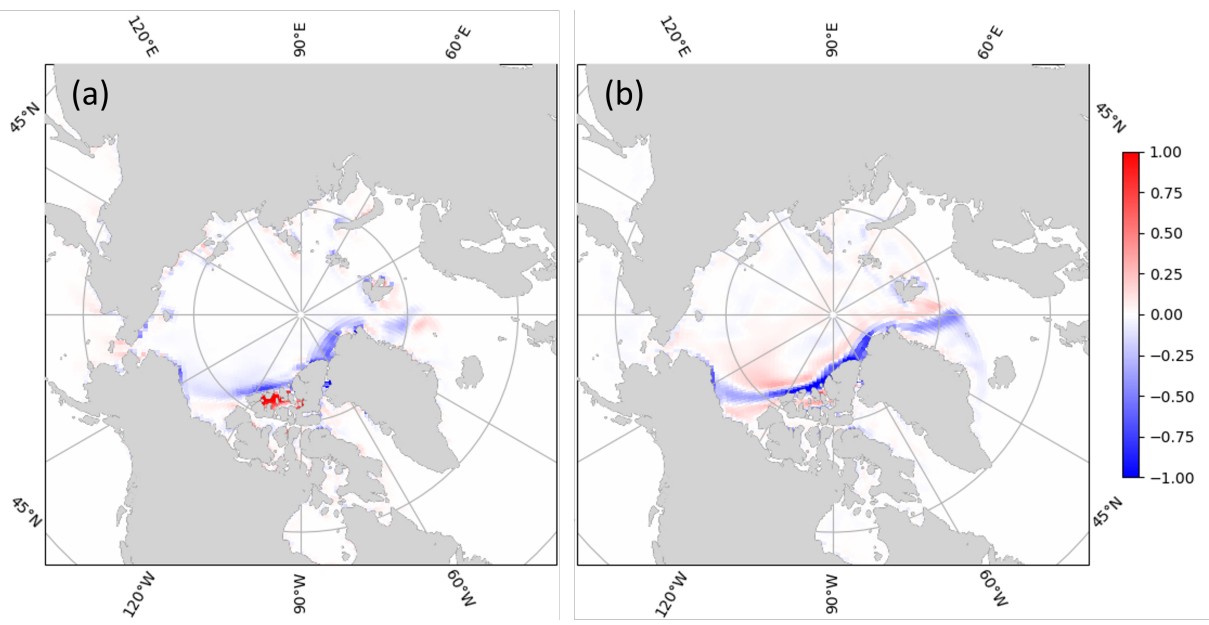

**Figure 9.** (a) Difference of the monthly mean sea ice thickness (m) after five years (December 2009) between a 1° C-grid simulation with remapping and a 1° B-grid simulation with remapping (reference). (b) Difference of the monthly mean sea ice thickness after five years (December 2009) between a 1° B-grid simulation with upwind and a 1° B-grid simulation with remapping (reference). This experiment is referred to as exp4 in Table 1.



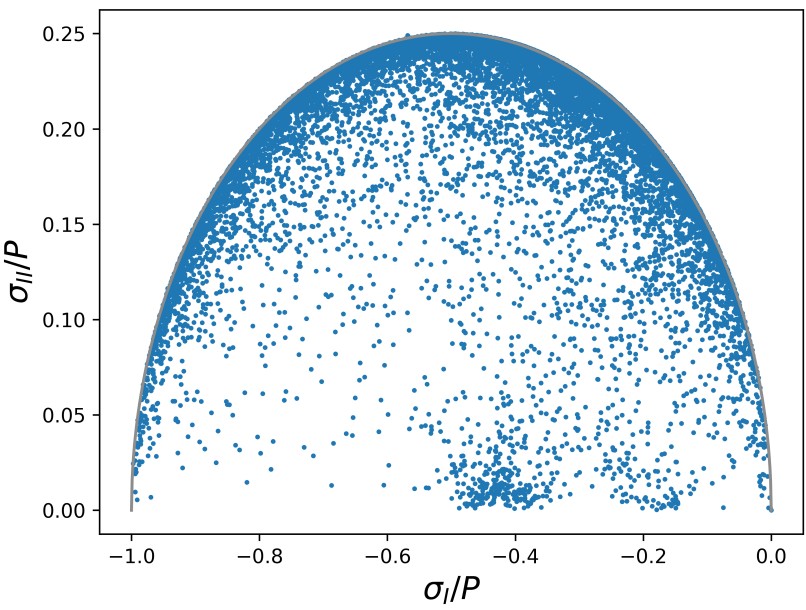

**Figure 10.** Stress invariants $\sigma_I$ and $\sigma_{II}$ normalized by the ice strength $P$. These are obtained from a snapshot after five days of a $1°$ C-grid simulation. This experiment is referred to as exp5 in Table 1.

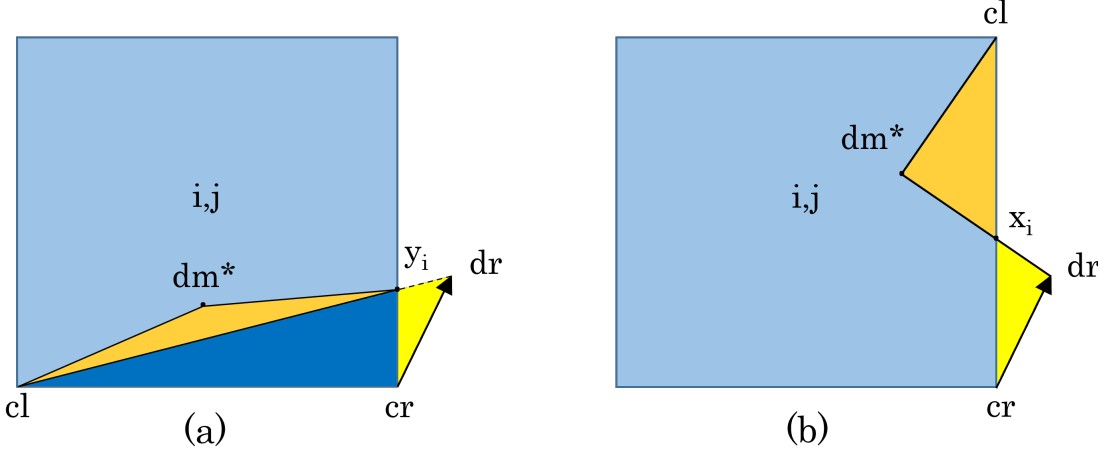

**Figure C1.** Schematic of departure regions on the south (a) and the east (b) edges of grid cell $i, j$ (in light blue). The same code is used to calculate the departure region for both edges. To do so, the nondimensional coordinate system is rotated by $90°$ for the east edge. This is why the corners for the east edge are also labeled as left ($cl$) and right ($cr$). The same convention applies to the departure points ($dr$). The orange triangles on both edges are defined by the EFA method by shifting the middle departure point to $dm^*$. $y_i$ is the intersection point on the $y$ axis for the south edge, and $x_i$ is the intersection point on the (rotated) $x$ axis for the east edge.



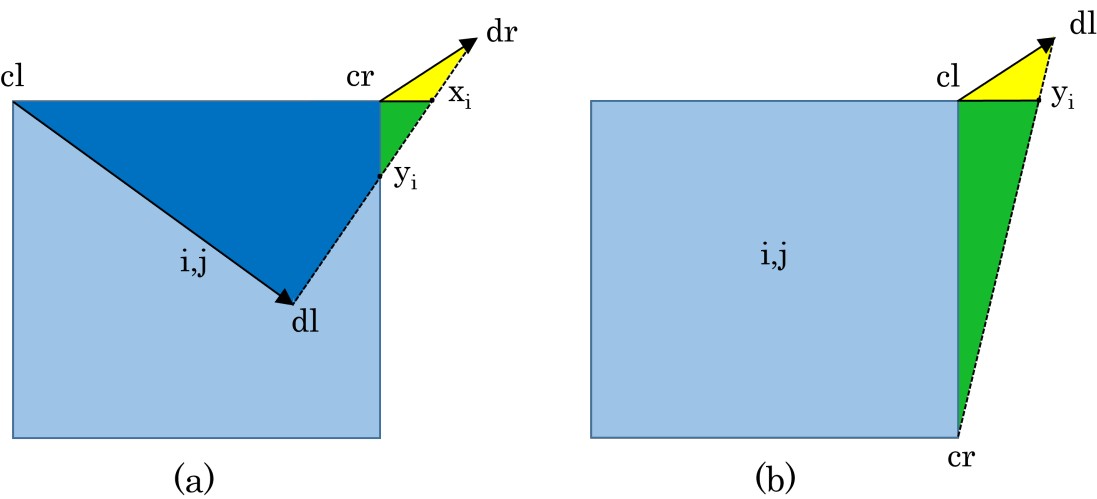

**Figure C2.** Schematic of departure regions on the north (a) and the east (b) edges of grid cell $i, j$ (in light blue). The same code is used to calculate the departure region for both edges. To do so, the nondimensional coordinate system is rotated by $90°$ for the east edge. The corners are labeled as left ($cl$) and right ($cr$). The departure points are $dl$ and $dr$, and $x_i$ and $y_i$ are intersecting points on the $x$ and $y$ axes.