# Peer review of "CICE on a C-grid: new momentum, stress, and transport schemes for CICEv6.5"

_Geoscientific Model Development, 2023_

## Author Comment (AC1)

**Response to reviewers' comments**

2 May 2024

We thank both reviewers for their time and their useful comments. Below, the comments from the reviewers are in black while our responses are in blue.

REVIEWER 1:

The manuscript describes a C-grid implementation of CICE. It is a well written paper, although I would question the advantages of the C-grid as compared to the variational implementation on the B-grid from purely numerical side – the implementation of rheology is of course possible on a quadrilateral C-grid, as demonstrated previously and in the present manuscript, but is not optimal in the sense that diagonal and off-diagonal components of the strain rate are naturally placed at T and U locations, while the field of Delta needs all them at the same location. Additionally, the variational (in essence bilinear finite-element) B-grid implementation is more accurate. Transport in narrow channels (not coupling, which is trivial) is a valid argument, but what is the accuracy of the divergence of internal stresses in such channels in a general case?

It would be interesting to compare simulated transport in a one-grid-cell wide channel with observations, but this is beyond the scope of the paper.

Leaving this discussion aside, I certainly recommend the manuscript as it be helpful for those users who prefer the C-grid implementation. The central place in the manuscript is devoted to the description of the adjustments of the incremental remapping methods used in CICE (EFA method), which is of obvious interest. It is, however, mentioned that this method was already available in CICE, which immediately leads to the question what is new? I would recommend to state more clearly what is the new part, and what is well known.

Our statement about the EFA method was incomplete, and did not accurately describe the contributions of this manuscript. The EFA method was developed by William Lipscomb and Mats Bentsen (co-authors on this manuscript), but the initial implementation in a previous CICE version was tested only for idealized cases and was never published. For this study, JF Lemieux and William Lipscomb analyzed and modified the scheme to make it robust for realistic sea ice simulations. It is a novel method, and it is fair to say that the EFA method

is a notable contribution of this work. We will make this clear in the revised manuscript.

I do not think the analysis presented in appendix B is relevant as it starts from a problematic assumption: for $u = u_0 + u' = u'$ as taken further in the analysis, (B1) is not reduced to $\sigma = -P$. Furthermore, the manuscript deals with the EVP method, and uses the replacement pressure, which is not reflected in the analysis, but is crucial for the behavior of perturbations. The answer is simpler. Generally one is not allowed to interpolate transport velocities, as the discrete divergence is only consistent in a certain sense. The Fourier symbol of C-grid divergence is $d = i[u \sin(ka/2) + v \sin(la/2)](2/a)$, where a is the cell side. The $u$ and $v$ velocities are not arbitrary, but given by the dynamics. Averaging to U-points and computing the divergence after will introduce $(1 + \cos(la))/2$ with $u$ and $(1 + \cos(ka))/2$ with $v$, i.e. will create a spurious additional divergence $d_{sp} = i[u \sin(ka/2)(\cos(la) - 1) + v \sin(la/2)(cos(ka) - 1)](1/a)$, which is simply a source of spurious modulation of scalar fields in (B7). The divergence $d$ in the experiment shown in Fig. 3(b) has a large scale structure, with the implication that the $u$ and $v$ contributions in $d$ nearly cancel for $k, l$ approaching to grid cutoff wavenumbers. But this would imply a substantial $d_{sp}$ at grid scales. This is a common issue, to avoid it one reconstructs scalars at faces or in control volumes, but not the advecting velocities. The EFA method restores the consistent divergence and removes spurious sources.

We thank the reviewer for carefully reading appendix B. We agree that assuming $u_0 = 0$ and that the ice is in convergence was not correct. However, we think our other assumptions were correct. The checkerboard pattern starts close to the wall where the ice converges. The convergence is then large enough such that the ice is in the plastic regime. We ran an experiment similar to the one used for Fig. 3a in the original manuscript. The EFA method is not used. The initial concentration is set to 1.0 so that the ice is strong right from the start of the simulation. We look for initial manifestation of the checkerboard pattern. After one day of simulation, the checkerboard pattern is observed in the concentration field, close to the eastern wall where the ice is being pushed (see Fig. 1 below). Where the checkerboard pattern is observed, the ice is in convergence (see Fig. 2 below). The high convergence values in the region of the checkerboard suggest that the ice is in the plastic regime. This is indeed the case if we look at the deformation $\Delta$ (see Fig. 3 below). $\Delta$ is notably larger than $\Delta_{min}$ in the region of the checkerboard, showing that the ice is in the plastic regime. Note that for previous time levels, the ice also converges and is in the plastic regime close to wall (not shown).

Because the ice is in the plastic regime, the replacement pressure is irrelevant and it is correct to assume that $\sigma = -P$ in the modal analysis. The problem is not related to the EVP method as the EVP does not lead to the checkerboard pattern when the EFA method is used.

Although $u_0$ is not zero, it is reasonable to assume it is small close to the wall due to the no-outflow boundary condition. Based on the experiment shown in Figs. 1-3 in this document, we estimated the four terms for the transport equation in our modal analysis. We find that $h_0 \sim 1$ m, $u_0 \sim 0.01$ m s$^{-1}$, $h' \sim 0.001$ m and $u' \sim 0.001$ m s$^{-1}$ such that the four terms are $u_0 h_0 \sim 10^{-2}$ m$^2$ s$^{-1}$, $u_0 h' \sim 10^{-5}$ m$^2$ s$^{-1}$, $u' h_0 \sim 10^{-3}$ m$^2$ s$^{-1}$ and $u' h' \sim 10^{-6}$ m$^2$ s$^{-1}$. This allows us to neglect the terms $u_0 h'$ and $u' h'$ and to end up with a similar modal analysis as in the original manuscript.

We ask the reviewer to look at our modified modal analysis in the revised manuscript. We are open to continuing the discussion with reviewer 1 and to further modifying our analysis if needed. We understand there are many assumptions in our modal analysis, but we like the fact that it considers the interaction between the momentum and transport equations.

Minor things:

line 22 'The C-grid ...' – but the B-grid may have some other advantages. I would prefer to see a more critical analysis here.

We agree. One advantage of the B-grid is that the $u$ and $v$ components are collocated, which simplifies the formulation of the Coriolis term and off-diagonal part of the water stress. One could also argue that the boundary conditions are simpler with the B-grid when calculating strain rates. This will be added in the revised manuscript.

line 31 - 40. I would say here that the manuscript presents an EFA version of the incremental remapping which is crucial for C-grid velocities. The history should be in the text, otherwise the story is told twice.

We agree. This will be mentioned in the revised manuscript.

line 50 ... (3) is hardly the main contribution – it is an element of history, the main contribution is the statement that the EFA method has to be used; As mentioned, I do not think the analysis (4) is rigorous enough to keep it in its present form in this manuscript.

Again, we agree with reviewer 1. We will clearly state in the revised manuscript that the EFA method is the main contribution and that it has to be used.

line 99 Why is $E_0$ less than 1? In the limit $dt-> 0$ this would require to increase the number of substeps.

The EVP method is a way to approximate the VP solution by adding an artificial elastic term. The idea is that at each time level, the elastic waves are

sufficiently damped such that the VP solution is obtained. The $E_0$ parameter defines the damping time scale (Hunke 2001). This is why it is smaller than 1.0 so that $E_0 \Delta t < \Delta t$ (i.e., the damping time scale is smaller than the time step).

line 205 ' mathematical mode' $->$ numerical mode

We agree. This will be modified in the revised manuscript.

Figure 5: In a way this illustration shows that the C-remap in not really an accurate scheme (although better than the first order upwind). How is it related to a slope or flux limited advection (e.g. with superbee)? Which accuracy is needed?

We would argue that the C-grid remapping is reasonably accurate, considering that the shape is advected for 30 days. It is our experience with the remapping that sharp features are diffused in the first few time levels because gradients are reduced by Van Leer limiting. After that, additional numerical diffusion is small, unlike first-order upwind methods in which strong diffusion continues.

line 325 Could the reason be that 'effective' diffusion associated with the advection is sufficiently high on coarse meshes?

We rather think that the checkerboard is not obvious in these global simulations because the wind forcing constantly changes direction and tends to smear out the checkerboard patterns. Thermodynamics probably also contributes to smearing out the pattern.

line 360 I think that the point is not in the null space created by averaging but in spurious divergence introduced by this averaging.

This is also related to the explanation of the checkerboard pattern. See our response above. Again, we are open to continuing the conversation with reviewer 1.

line 450 Formulas below are written under an assumption that $\xi_1$ and $\xi_2$ parameterize a flat manifold. In reality, a covariant derivative of velocities will contain a vertical contribution on the sphere (try to compute a vector Laplacian in the same way as A13-A16 to see that some terms are lost)

Ok. This will be mentioned in the revised manuscript.

line 611 For the velocity field used further $D_d$ is an oscillating function, which makes this statement contradictory.

We agree this was incorrect. See our response above about our modified modal analysis.

753 Why 'in CICE'? This is a general expression.

We agree. This will be modified in the revised manuscript.

REVIEWER 2:

The paper describes the implementation of a C-grid discretization in CICE. It explains how the C-grid discretization can be coupled with an incremental mapping scheme originally developed for a B-grid. The authors follow the approach of Kimmritz 2018 and Bouillon 2013 in implementing the C-grid for the momentum equation. The main part of the paper describes how the existing incremental remapping scheme in CICE can be modified to work with the C-grid approximation of the momentum equation. To my understanding, the coupling (modification) of the remapping is the only scientifically new development/method that is not documented in another paper. However, the coupling presented is not discussed in relation to existing approaches in the field/literature. It seems inconsistent to me to implement a C-grid for the momentum equation, but adapting a B-grid transport (instead of using C-grid schemes e.g., the second order DG, which is more accurate). Furthermore, the proposed EFA scheme has problems on non-uniform grids.

We agree that a better description of existing approaches is needed. We will include a more complete literature review with better description of other models and methods in the revised manuscript.

We disagree with the reviewer about the lack of novel ideas presented in this manuscript. First, as replied above to the other reviewer, the EFA method was never introduced and described before in a paper. The method was developed by William Lipscomb and Mats Bentsen (co-authors on this manuscript) and modified more recently by JF Lemieux and William Lipscomb to increase its robustness. The EFA method also allows transport in one-grid cell wide channels. As such, the remapping with the EFA method should be seen as a hybrid B- and C-grid transport scheme. We think these are notable contributions that should be shared with the community. We also disagree that "the proposed EFA scheme has problems on non-uniform grids." The earlier problems have been fixed, as thoroughly described in the manuscript (Appendix C).

We also reiterate that we have developed a novel analytical solution for one-grid-cell channels. To our knowledge, this has never been published before. Due to the complexity and nonlinearity of the sea ice momentum equation (with a VP formulation), analytical solutions are difficult to obtain. This new analytical solution has been very useful for validating the implementation of the VP rheology on a C-grid, and could certainly help others model developers.

Figure 5 shows the transport in a one cell wide channel and compares the C-upwind to the C-remap. Both approaches are diffusing. There are also DG type transport schemes which work in one cell wide channel. These methods are able to preserve the shape e.g. The neXtSIM-DG dynamical core: A Framework for Higher-order Finite Element Sea Ice Modeling. I think this should be discussed and acknowledged in the paper.

Yes, there is numerical diffusion in the C-remap solution, but as we argue above, the remapping solution is reasonably accurate considering the sharp initial gradients and the 30-day advection period. It would be interesting to compare the remapping-EFA solution to a DG approach, but this is beyond the scope of this manuscript.

The first reason for using remapping is to reuse part of the code. This, of course, simplified the implementation of a C-grid version of CICE. This choice was also motivated by many nice characteristics of remapping: the scheme is stable, conservative, preserves monotonicity (and compatibility of associated tracer transport) and computationally efficient when multiple thickness categories and tracers are transported (Lipscomb and Hunke 2004).

Remapping is efficient for many tracers and thickness categories because most of the computations are geometrical; they are performed once per time level instead of being repeated for each tracer and thickness category. It is written, in the paper proposed by the reviewer, that "The parallel efficiency of the advection scheme is not as good as the efficiency of the mEVP iteration. In this benchmark problem, this is not significant since only two tracers are advected and since there are 100 substeps of the mEVP solver in each advection step. For more complex thermodynamics, the situation will be different and further optimizations appear to be necessary." Hence, even if DG were less diffusive than remapping, it might not be as computationally efficient for a large number of tracers. This would need to be investigated in another research project.

We agree, however, that DG and other transport schemes should be discussed. We will do this in the revised manuscript.

The authors state several times that the most difficult part is the approximation of the rheology term. But how well are the stresses and the strain rates approximated compared to the existing B-grid approach in CICE? To address the issue, I would solve a 2D equation

$$div(\nabla v + \nabla v^T) = f$$

calculate the analytical solution for this and check under mesh refinement how the error convergences for both approaches, C-grid and B-gird. When it comes to the fully coupled system how large the coupling error? Can you represent LKFs with the same quality with the C-grid?

This is beyond the scope of the paper. We do not investigate the ability of the B and C-grid discretizations to represent linear kinematic features. This was studied by Melhmann et al. (2021), who concluded that the B- and C-grid discretizations have similar abilities to simulate these features.

Minor things:

Introduction: I miss a general overview of how other models with C-grid deal with the advection and how they couple to the advection. For example, the MITgcm. Is your proposed coupling more or less accurate?

We agree that the manuscript should give an overview of what is used in other models. This will be added in the revised manuscript.

Introduction: Could you give more details on the EFA method. Who developed the EFA method? Is it used in other research fields?

We agree that the presentation of the EFA method in the first version of the manuscript was incomplete. The manuscript did not convey that the EFA method is a novel approach that has not been published before. The method was first developed by William Lipscomb and Mats Bentsen (co-authors on this manuscript) and was modified more recently by JF Lemieux and William Lipscomb to increase its robustness. It will clearly be stated in the revised manuscript that this is a scientific innovation.

L.144 What is the difference to Bouillon and Kimmritz? If there is no important difference why do repeat it in the appendix?

The manuscript not only describes our scientific innovations (EFA method, new analytical solution) but also will serve as a reference and guide for users of CICEv6.5. This is why the manuscript provides more details than the Bouillon and Kimmritz papers. Note that the treatment of the seabed stress term is also presented. This term was not included by Bouillon and Kimmritz.

L 213 " Fortunately" It sounds like you are happy to avoid programming. I would remove the word.

See our other comments above. This will be rephrased in the revised manuscript.

166 How much accuracy do you lose by the averaging?

Our current finite-difference C-grid implementation requires the computation of the viscous coefficients at the $U$ point. Ultimately, the question is how much accuracy is lost for the simulated velocity. This is a difficult question and

beyond the scope of this paper. Intuitively, the largest errors are probably where there are strong gradients in the ice strength and in deformations...basically where there are LKFs.

257 How does the EFA method ensure that the divergence is consistent with the divergence calculated by the dynamical solver?

Given the C-grid east edge velocity $u_E(i-1,j)$, the EFA method adjusts the departure region to ensure that the total flux is equal to $u_E(i-1,j)\Delta y_E\Delta t$. This is done on the four sides.

343 Here are too many dots

This will be fixed in the revised manuscript.

381-3.85 I think it is trivial that this is fulfilled since your using C-grid for the momentum equation that has already been used by other models.

It is not because we followed what others implemented that we did it correctly! This is just a useful validation. It is often used to verify the implementation of the VP rheology (e.g. Geiger et al. 1998, Hunke 2001, Bouillon et al. 2013, Mehlmann and Richter 2017)

Jean-Francois Lemieux

[Figure]

Figure 1: Ice concentration after 1 day of simulation. In order to see the checker-board, the range is between 0.9994 and 1.0.

[Figure]

Figure 2: Snapshot of divergence after 1 day of simulation. The colors (other than dark red) show regions of convergence. Values are capped below at $10^{-7}s^{-1}$.

[Figure]

Figure 3: Log10 of the deformation $\Delta$ used to calculate the viscosity.

---

## Author Response (AR2)

**Response to reviewer's comments**

6 July 2024

We thank the reviewer for his/her time and professionalism. Below, the comments from the reviewers are in black while our responses are in blue.

REVIEWER 1:

I am satisfied with the answers and the revision made. Below are some minor things and some answers or discussion related to the questions raised previously. The manuscript can be accepted after they are addressed. The manuscript needs not to be sent to me another time.

23 water stress?

As most ocean models nowadays represent the Ekman spiral, there is usually no off-diagonal term associated with the water stress. However, when a non-zero turning angle is used, there is an off-diagonal term. This can be seen in equations A46 and A47.

41 Is not piecewise-constant equivalent to the first-order upwind?

We have clarified in the revised manuscript (lines 40-42) that the upwind scheme in SIS2 is based on a non-directionally split approach while the other options are directionally-split piecewise parabolic, directionally-split piecewise linear, and directionally-split piecewise constant methods.

55 'incremental remapping is much less diffusive...' – it is not so the remapping, but linear reconstruction in cells, which makes the method higher-order. Without it the method would be very similar to upwind.

Good point. We added some text in the revised manuscript (lines 55-57) about the linear reconstruction of scalar fields.

59 Noise is mentioned here for the first time, and your reader is unaware about it, so provide some details.

> We agree. We added a paragraph about this in the introduction of the revised manuscript (lines 59-62).

66 'description ...' – I would say about interpretation and elimination.

> We have replaced 'description' by 'interpretation' (line 72 in the revised manuscript). We refer to the 'elimination' of noise a few sentences above.

75 Appendices B1-... (if you use plural).

> Corrected.

114-115 Keeping $E_0$ small would require to increase $n_{evp}$ on finer meshes. Instead, one could fix $T_d$ independent of model step, assuming that the convergence to VP will be achieved during several time steps.

> We agree this would be a possibility but it would need to be tested. We prefer not to mention this here as this is beyond the scope of this work.

395-398 Do you need this in Conclusions? This is just sanity check, and there is no ground to expect something different.

> We agree this is not needed. We have removed the paragraph in the revised manuscript.

Two questions related to the discussion in the first review.

1. Divergence noise: The explanation proposed by the authors can be a part of the story. It, however, contains too many assumptions, such as for example $u_0$ is considered to be small (and $u'$ is nevertheless kept), and fluctuations of concentration are ignored although they will be appearing with a large factor about 20 in the linearization. Furthermore, the mode it identifies is the mode in divergence, and it is seen immediately. For a Fourier mode, C-grid divergence with respect to T point is $(2/h)(u_E \sin(kh/2) + u_N \sin(lh/2))$. If velocities are interpolated to B-grid, the divergence becomes $(2/h)(u_E \sin(kh/2) \cos(kh/2) + u_N \sin(hl/2) \cos(lh/2))$. (i) It has a mode. (ii) it is in error even if $kh$ is small, but $lh$ is not, or vice versa. While I do not have much against the analysis in the manuscript, it is not really needed. If there is a grid-scale component in velocity, it will be decoupled from the grid-scale component in tracers, which results in non-propagating perturbations.

> Event though our interpretation relies on a simplified set of equations we would prefer to keep it. We already mentioned that many assumptions have to be made for our modal analysis. Second, we mention that this is our interpretation of the checkerboard pattern. We also added the following sentence (lines 682-683), in line with the reviewer's comments, in Appendix B: "This

$[1 + \cos(l\Delta y)]$ term characterizes the spurious divergence associated with the interpolation of velocities to the $U$ points."

2. The contributions from metric factors: The authors are right saying that metric factors are everywhere treated as in the manuscript. However, take, spherical geometry. The $r\phi$ and $r\theta$ components will generally appear in strain rate tensor even if velocities are tangent to the surface. Their divergence with respect to $\phi$ and $\theta$ will contribute to tangent vectors. The well-known expression for vector Laplacian in spherical coordinates includes such contributions ($-u/\cos^2(\theta)/R^2$ instead $-u\sin^2(\theta)/\cos^2(\theta)/R^2$ which will be obtained if they are ignored). I do not think further discussion is needed, since now the approximation is mentioned in the manuscript.

Again we would like to thank the reviewer for his/her many useful comments and critiques.

Jean-Francois Lemieux